# Effect of Dietary Cold-Pressed Hempseed Cake Supplemented with Tomato Waste on Laying Hen Performance and Egg Yolk Lipid Profile and Antioxidant Status Before and After Storage

**DOI:** 10.3390/ani14233444

**Published:** 2024-11-28

**Authors:** Daniel Mierlita, Stelian Daraban, Alin Cristian Teușdea, Alina Stefania Stanciu

**Affiliations:** 1Department of Animal Nutrition, Faculty of Environmental Protection, University of Oradea, 1 University St., 410087 Oradea, Romania; daniel.mierlita@uoradea.ro (D.M.); ateusdea@gmail.com (A.C.T.); 2Department of Technological Science, Faculty of Animal Science and Biotechnologies, University of Agricultural Sciences and Veterinary Medicine, 3-5 Manastur St., 400372 Cluj-Napoca, Romania; ovineusamv@yahoo.com

**Keywords:** hempseed cake meal, tomato waste, fatty acids, antioxidant status, egg yolk, laying hens

## Abstract

Phytochemicals in tomato waste are known for their high antioxidant activity, and carotenoid pigments can be harnessed as natural egg yolk pigments. Also, hempseed cake (HSC) has been recognized as a valuable source of protein and essential fatty acids for poultry nutrition. However, few studies have investigated the use of tomato waste in the diet of laying hens and its effect on egg quality traits and the oxidative stability of egg yolk enriched in polyunsaturated fatty acids (PUFAs). In the present study, including hempseed cake (HSC) in the diet of laying hens as a source of PUFAs, along with tomato waste (dried whole tomato, DT, or dried tomato pomace, DTP), improved the yolk color and the content of antioxidant compounds and omega-3 fatty acids in the egg yolk. In addition, the dietary incorporation of tested byproducts reduced the n-6/n-3 FA ratio, hypercholesterolemic FA and cholesterol content of egg yolk and improved the oxidative stability of the yolk during egg storage. In conclusion, in this study, we showed the high antioxidant potential of tomato waste, especially DTP, and its potential use in enriching the diet of laying hens with PUFAs with the aim of improving the nutritional quality of eggs and contributing to the promotion of human health and also to environmental protection and the reduction in waste from the food industry.

## 1. Introduction

Omega-3 fatty acids (n-3 FAs) play an important role in reducing the incidence of cardiovascular disease, inflammation and diabetes [1]. Because the human body cannot synthesize n-3 FAs, the fortification of table eggs with n-3 FAs has been shown to be a successful strategy to ensure the adequate intake of n-3 FAs in the human diet. Numerous studies have shown that chicken eggs can be enriched in n-3 FAs by supplementing the diets of laying hens with sources rich in n-3 FAs, such as fish oil, seaweed, oil or byproducts of flax, hemp, camelina and rapeseed [2,3,4]. Recent research has shown that hemp seeds and byproducts, such as hempseed cake resulting from cold pressing of the seeds, can be used in the diet of laying hens, especially as a source of n-3 FAs [5,6,7]. These studies revealed an increase in n-3 FA content in the egg yolk and a reduction in the n-6/n-3 FA ratio, indicating the increased nutritional quality of table eggs.

However, n-3 FAs, especially long-chain FAs [i.e., eicosapentaenoic acid (EPA, C20:5n-3) and docosahexaenoic acid (DHA, 22:6n-3)], are highly susceptible to oxidation because of their double bonds, which cause eggs enriched in n-3 FAs to deteriorate faster and have a shorter shelf life [2,3,6]. A study by Eid et al. [2] indicated that the incorporation of fish oil into the diet of laying hens at a level of 5% to obtain n-3 FA-enriched eggs may lead to increased lipid peroxidation and decreased antioxidant activity. To avoid or mitigate the yolk lipid peroxidation process, especially during egg storage, it is necessary to incorporate feed additives with an antioxidant role in the diet of hens. An alternative to synthetic additives with an antioxidant role used in poultry feed could be plants and food wastes rich in antioxidant compounds. Natural antioxidants have the ability to inhibit oxidation processes and thus contribute to the maintenance of egg quality during storage [8]. Dietary supplementation with natural sources of antioxidants has been shown to increase egg yolk antioxidant content and delay lipid oxidation in egg yolk enriched in unsaturated FAs [4,9].

Substantial amounts of tomato waste are produced annually worldwide due to the overproduction of fresh tomatoes and the industrial processing of tomatoes [10]. It has been estimated that about 31% of all fresh, marketed tomatoes end up as waste, while about 30–40% of industrially processed tomatoes end up as “waste” commonly called tomato pomace, consisting mainly of skin and seeds [11]. Tomatoes contain large amounts of lycopene (80–90% of total carotenoids) and β-carotene (7–10%) and smaller amounts of lutein, phenols, tocopherols, zeaxanthin and vitamin C, which exhibit strong antioxidant action [12,13]. These phytochemicals can accumulate in egg yolk and improve the oxidative stability of yolk lipids [12]. In addition, carotenoid pigments from tomato waste can improve egg yolk color by replacing color additives in hen feed.

A study carried out by Correia et al. [14] demonstrated that the use of dried tomato pomace (DTP) in high-fat pig diets increased the tocopherol concentration and oxidative stability of the meat. Rezaei and KazemiFard [15] demonstrated that the inclusion of 4% and 12% tomato powder in the diet of Japanese quail improved the oxidative stability of eggs by reducing the concentration of malondialdehyde (MDA, a byproduct of lipid oxidation) in the yolk. In laying hens, the inclusion of DTP in the diet at a level of 6% has been shown to be beneficial for yolk color, lycopene concentration and reduced yolk cholesterol content [16]. De Souza Loureiro et al. [17] observed a reduction in egg production and egg weight when DTP was introduced into the diet of hens at a level higher than 5%, while other studies [18] reported that the incorporation of DTP in hen feed at a level of 20% did not influence egg production or egg weight.

Few studies have explored the effect of using tomato waste enriched in unsaturated FAs in the diet of laying hens [13,19]. In addition, to the best of our knowledge, this is the first study to compare the nutritional and functional properties of dried whole tomato (waste resulting from the sale of tomatoes for fresh consumption) with dried tomato pomace and its effect as a source of bioactive compounds in laying hen feed containing a rich source of PUFAs. Therefore, the aim of this study was to evaluate the effects of dietary inclusion of HSC as a natural source of polyunsaturated FAs together with a natural source of antioxidants such as DT or DTP on the performance of laying hens and egg quality, as well as the fatty acid profile, phytochemical content and oxidative stability of eggs enriched in n-3 FAs before and after storage. The findings of this study will help to improve the formulation of diets for laying hens to produce eggs fortified with n-3 FAs and natural antioxidants, using industrial food byproducts such as hempseed cake and tomato waste.

## 2. Materials and Methods

### 2.1. Ethical Approval

The experimental procedures were approved by the Ethics Committee of the Faculty of Environmental Protection of the University of Oradea (protocol code 6/04/11/2024) in accordance with the legislative regulations (Law 206/2004, Directive 2010/63/EU, Law 43/2014) on the care and use of animals for scientific purposes.

### 2.2. Experimental Materials

Hempseed cakes were purchased from a local (Salonta, Romania) producer who obtains hemp oil from cold-pressed hemp seeds.

Whole fresh tomatoes rejected for market sale (overripe, cracked tomatoes, with defects in shape and size, etc.) were purchased from local producers and local markets (Oradea, Romania), and the tomato pulp was supplied by Food Transylvania, a fruit and vegetable processing plant in Romania. The tomatoes were cut into small pieces (the spoiled portions were removed), after which they were dried in a static oven at 50 °C together with the tomato pomace for 24 h and then ground using a universal hammer mill with 1 mm mesh.

Tomato and HSC scraps were vacuum packed in dark foil bags and stored in a cold room at 4 °C until use. Before being introduced into the feed of laying hens, HSC and tomato waste were analyzed in the laboratory for proximate composition, FA profile, antioxidant profile and ABTS radical-scavenging activity, expressed as TE (Trolox equivalent).

### 2.3. Experimental Design

A total of 96 TETRA SL laying hens during the peak laying period (28 to 37 weeks of age; initial body weight: 1830 ± 77 g), when egg production is relatively constant, were divided into three groups of 32 birds each (8 replicate cages, 4 hens per cage). The birds were randomly assigned to three dietary treatments: a standard corn–soybean meal diet (C), a diet containing 20% hempseed cake and 4% dried whole tomato (HT) and a diet containing 20% hempseed cake and 4% dried tomato pomace (HTP). The feeding trial lasted 10 weeks.

In the present study, HSC was used to enrich the egg yolks with n-3 FAs at an inclusion level of 20% in the hens’ diet in agreement with a previous study [6]. DT and DTP were included in the diet of laying hens at a level of 4%, along with HSC, to serve as complementary sources of natural antioxidants. Previous studies [15,16,17] have shown that supplementing the diet of laying hens or Japanese quail with tomato waste at a level of 4% to 6% improved the yolk color and yolk content of antioxidant compounds.

Hens from each replicate were allocated to a common cage (60 × 60 × 40 cm) so that feed intake and egg production could be recorded separately. Each cage was provided with a feeding trough and two water nipples. Access to feed and water was provided ad libitum.

All groups benefited from similar management conditions and a controlled microclimate (temperature: 21–22 °C, humidity 65–68%). The birds had a 16 h light schedule (photoperiod from 06:00 to 22:00).

### 2.4. Dietary Treatments

Two weeks before the start of the experiment, all three groups of hens were fed a commercial diet to allow them to adapt to the new conditions and achieve standard egg production.

The tested diets (C, HT and HTP) were formulated to be isocaloric and isonitrogenous. To ensure the same content of CP (crude protein) and ME (metabolizable energy) in all diets, in the standard diet (C), we incorporated sunflower meal and sunflower oil, respectively (Table 1). Diets were formulated using HYBRIMIN^®^ Futter 5 software to meet the feeding requirements of laying hens [20]. Diets were prepared fresh every 2 weeks and stored in a cold room.

### 2.5. Performance Parameters

The body weight of the hens was recorded at the beginning and end of the experiment. Feed intake was measured weekly by weighing and subtracting the remaining feed. Eggs were collected by hand every day, and twice a week, the eggs were weighed individually. Feed intake, the number of eggs and their weight were recorded separately for each replicate (n = 8 replicates/treatment). Egg mass was calculated based on laying rate and egg weight.

### 2.6. Egg Sampling

In weeks 5 and 10 of the experimental period, 32 eggs/treatment (4 eggs/replicate) were taken to determine the quality parameters of fresh eggs (16 eggs/treatment) and eggs stored for 30 days at 4 °C (16 eggs/treatment). The weight of the albumen, yolk and shell; shell thickness; and yolk color were determined. Egg component weights were determined by weighing using an electronic scale (Mettler-Toledo LLC, Columbus, OH, USA). Peel thickness was measured with a Mitutoyo digital micrometer (Kawasaki, Japan) at three locations (point, equator and air cell), and we used the average value. The yolk color was measured using the 15-point Roche scale (color scale from 15, dark orange, to 1, light pale).

In the ninth week of the test, 64 eggs were collected per dietary treatment (8 eggs per replicate for each diet), of which 32 eggs/treatment were processed and analyzed as fresh eggs, and the remaining 32 eggs/treatment were stored at 4 °C for 30 days. These eggs were broken and the yolk was separated, and then the yolks of two eggs from the same replicate were combined, thus obtaining 32 yolk samples per treatment (16 samples for fresh eggs and 16 samples for stored eggs). The yolks, protected from light with aluminum foil, were frozen at −80 °C and later analyzed to determine the FA profile, the antioxidant profile (lycopene, lutein, β-carotene, retinol, α-tocopherol and total phenols) and the oxidative stability of yolk lipids (Trolox equivalent antioxidant capacity and MDA concentration).

For yolk cholesterol analysis, 24 eggs/treatment (3 eggs/replicate) were sampled during the last week of the experiment.

### 2.7. Chemical Analyses

#### 2.7.1. Proximate Chemical Composition of Feed

Hempseed cake (HSC), dried whole tomato (DT), dried tomato pomace (DTP) and test diets (C, HT and HTP) were analyzed according to AOAC [22] methods for DM (gravimetric method; method 934.01), crude protein (CP) (N × 6.25; method 954.01; Kjeltec auto 1030; Tecator Instrumente, Höganäs, Sweden) and crude fat (EE) (petroleum ether extraction; method 920.39; SOXTHERM, C. Gerhardt GmbH, Königswinter, Germany). Neutral detergent fiber (NDF) and acid detergent fiber (ADF) contents were determined according to the method described by van Soest et al. [23] using the ANKOM 220 analyzer (ANKOM Technology Corporation, Fairport, NY, USA). The contents of starch (STA) and sugar (SUG) in HSC, DT and DTP were determined [22], and the content of AMEn (apparent metabolizable energy corrected for nitrogen balance) was calculated. The AMEn of HSC, DT and DTP was estimated according to the equation proposed by the World’s Poultry Science Association [24] as follows:AMEn (kJ/kg DM) = 15.51 (CP) + 34.31 (EE) + 16.59 (STA) + 13.01 (SUG).

All chemical analyses were performed in triplicate.

#### 2.7.2. Yolk Cholesterol Content

Yolk cholesterol was determined according to AOAC (method 994.10) using a Perkin-Elmer GC (Shelton, MA, USA), as previously described [4]. Briefly, after saponification, the yolk sample was treated with petroleum ether and brought to neutral pH with distilled water. After treatment with chloroform, the obtained extracts were injected into an HP-5 GC fused silica capillary column (30 m × 0.32 mm id, 0.1 µm film thickness, Agilent J&W GC Columns, Santa Clara, CA, USA) and analyzed on a flame ionization detector (FID). Cholesterol was identified using the laboratory standard solution, and the cholesterol content was expressed as mg/g yolk.

#### 2.7.3. Feed, Diet and Egg Yolk Fatty Acid Analysis

Fatty acids (FAs) were determined as methyl esters via GC/MS using a gas chromatograph (Shimadzu GC-2010 Plus, Tokyo, Japan) equipped with an FID and an HP-88 column (100 m × 0.25 mm ID), as previously described [4]. External standards (Supelco 37 Component FAME mix; Supelco Bellefonte, PA, USA) were used for the identification of fatty acids, and the results were expressed as % FA of total FAs.

#### 2.7.4. Determination of Antioxidant Compounds

Carotenoids were determined using high-performance liquid chromatography (HPLC) according to the procedure described by Schlatterer and Breithaupt [25]. Extraction was performed with a mixture of methanol/ethyl acetate/petroleum ether (1:1:1, *v*/*v*/*v*) via centrifugation (6 min at 6000 rpm) until the supernatant became colorless. The supernatant was treated with NaCl solution, and the upper layer was collected, dried and then dissolved with acetonitrile/methanol/ethyl acetate (3:1:1, *v*/*v*/*v*) and filtered through a membrane of 0.45 µm. HPLC analyses were performed on a Shimadzu LC20 AT system with an SPDM20A diode array detector. The detector was set to 450 nm. Standards for lycopene, lutein and β-carotene were obtained from Sigma-Aldrich (Steinheim, Germany). To compare the peak areas of carotenoids with those of specific standards, we used Chrom Quest 4.2 software.

Retinol and α-tocopherol in feed and yolk samples were quantified using an HPLC device equipped with a PDA-UV detector (Finningan Surveyor Plus, Thermo-Electron Corporation, Waltham, MA, USA) at 325 nm for retinol and 292 nm for α-tocopherol, as previously reported [4].

The total phenolic content (TPC) of the feed ingredients (HSC, DT and DTP) and egg yolk was determined using the standard Folin–Ciocalteu method as previously described [4]. Briefly, 0.8 mL of deionized water, 0.1 mL of Folin–Ciocalteu reagent and 0.5 mL of Na_2_CO_3_ solution (30%) were added to 0.1 mL of methanolic extract. After 30 min of incubation at room temperature, the samples were stored for one hour in the dark. The absorbance of the mixture was measured at 750 nm in an automatic reader (Infinite 2000, Tecan, Salzburg, Austria). A gallic acid standard curve was obtained, and the results were expressed in mg gallic acid equivalents (mg GAE/g).

#### 2.7.5. ABTS Radical-Scavenging Assay

The antioxidant activity of the tested food waste (HSC, DT and DTP) and fresh and stored egg yolk was evaluated with a 2,2-azino-bis(3-etilbenzotiazolin)-6-sulfonic acid (ABTS) assay, according to the procedure described by Mierlita et al. [4]. The ABTS radical-scavenging assay is used to measure the capacity of compounds to reduce the dark blue ABTS•+ radical cation to colorless ABTS, which can be quantified spectrophotometrically. Briefly, a mixture of ABTS with potassium persulfate (1:1, *w*/*w*) was kept in the dark (12 h) and then diluted with ethanol to an absorbance of 0.7 ± 0.05. The homogenized samples were treated with the obtained mixture, and after 30 min, the absorbance of the mixture was measured at 734 nm. The results were expressed as μmol Trolox equivalents (TE)/g of sample.

#### 2.7.6. Oxidation Assessment

Thiobarbituric acid reactive substances (TBARSs) were used as markers of lipid oxidation in egg yolk and expressed as malondialdehyde (MDA) equivalents as previously described [4]. The method is based on the reaction between MDA and thiobarbituric acid (TBA), which forms a pink-colored complex. The absorbance of the solution was determined at 532 nm using a Hitachi U-2000 spectrophotometer (Hitachi, Ltd., Tokyo, Japan). The results were expressed as µg MDA/g yolk after the preparation of a standard curve of MDA tetrabutylammonium salt (Sigma-Aldrich, Buchs, Switzerland).

### 2.8. Estimation of Lipid Health-Related Quality Indices

The average concentration of each fatty acid was used to calculate the amounts of FAs (SFA, UFA, MUFA, PUFA, hypercholesterolemic FA and hypocholesterolemic FA), qualitative indices, nutritional indices and metabolic indices.

Based on the lipid profile of the egg yolk, qualitative indices (PUFA/SFA, n-6/n-3 FA, LA/ALA, saturation index (SI), peroxidability index (PI), oxidative susceptibility (OS), unsaturation index (UI)), nutritional indices (nutritional value index (NVI), atherogenic index (AI), thrombogenic index (TI), egg lipid quality index (LQI), ratio of hypocholesterolemic and hypercholesterolemic FA (h/H FA), health-promoting index (HPI), desirable FAs (DFAs)) and metabolic indices (elongase index (EI), thioesterase index (ThI), ∆9-desaturase C16:1+C18:1, activity index) were determined using the appropriate formulas [9,26,27].

### 2.9. Statistical Analysis

Data were subjected to one-way ANOVA (analysis of variance) for three treatments and eight replicates with four hens in each replicate by using the general linear model (GLM) procedure in SAS software [28] for a completely randomized design. The data on the laying performance of the hens and lipid health-related quality indices in the yolk were tested only for the effect of diet, and the other data (egg quality traits, yolk FA profile, yolk antioxidant profile and oxidative stability of yolk lipids) were tested for “type of diet” (C, HT and HTP) and “type of eggs” (fresh or stored). The treatment means were compared using Duncan’s multiple range test, and statistical significance was set to *p* < 0.05.

Correlations between some yolk characteristics and the Kruskal–Wallis (*p* = 0.05) non-parametric test followed by multiple comparisons (Dunn test, *p* = 0.05) were calculated with Stata 17.0 SE (StataCorp LLC, StataCorp, 4905 Lakeway Drive, College Station, TX, USA). A program developed in MATLAB 2023a 9.14.0 CWL (The MathWorks Inc., 1 Apple Hill Drive, Natick, MA, USA) was used for principal component analysis (PCA).

## 3. Results

### 3.1. Chemical Compositions of Hempseed Cake, Tomato Waste and Experimental Diets

Tomato waste samples showed a high moisture content, being higher (*p* < 0.001) for whole tomatoes (952.6 ± 4.2 g/kg) than for tomato pomace (691.8 ± 3.1 g/kg) (Table 2).

Hempseed cakes obtained via cold pressing proved to be a good source of proteins (32.21% of DM) and also of fats (14.32% of DM), the latter contributing to the high energy value of HSC (AMEn: 2830 kcal/kg DM). A considerable amount of fat (11.83% of DM) was also observed in DTP (Table 2).

The contents of crude fiber, NDF and ADF were higher (*p* < 0.001) for DTP than for HSC and DT (DTP > HSC > DT), which limits their use in large quantities in poultry feed.

Lycopene was the dominant component in the carotenoid profile of tomato waste (DT and DTP), followed by β-carotene and lutein. The concentration of lycopene and β-carotene was higher (*p* < 0.001) in DTP than in DT. The total carotenoid content was 550.8 mg/kg in DT and 1082.3 mg/kg in DTP, almost two times higher in DTP than in DT. In addition, DTP had higher concentrations total phenols and α-tocopherol and also a higher ABTS radical-scavenging activity compared to DT and HSC (*p* < 0.001) (Table 2).

The results presented in Table 3 show that HSC is a major source of PUFAs, especially linoleic acid (56.38% of total FA) and α-linolenic acid (20.91% of total FA). The ALA content in DTP and DT was almost seven times lower than that in HSC (*p* < 0.001). Consequently, the best n-6/n-3 FA ratio was found in HSC, being almost five times lower than that in DT and DTP (*p* < 0.001). The FA profile showed no differences (*p* > 0.05) between DT and DTP.

The dietary incorporation of the tested byproducts (HSC, DT and DTP) led to higher determined concentrations (*p* < 0.001) of n-3 FAs in hen feed, being 3.9 times higher in the HT diet and 4.6 times in the HTP diet than in the control diet. On the other hand, the concentration of n-6 FAs was higher in the fat of the C diet than in the HT and HTP diets, and consequently, the dietary ratio of n-6/n-3 FAs was 4.3 times lower in the HT diet and 4.8 times lower in the HTP diet than in the C diet (Table 3).

### 3.2. Performance of Laying Hens and Egg Quality

The effects of the experimental diets on the performance and physical characteristics of the eggs before and after storage are summarized in Table 4 and Table 5.

The dietary inclusion of HSC in combination with tomato waste (DT or DTP) increased daily feed intake (*p* < 0.05) compared to that in the control group, while daily egg production, average egg weight and feed conversion ratio (FCR) were not affected by dietary treatments (*p* > 0.05) (Table 4). No significant differences were found between the diet supplemented with DT (HT group) and the diet supplemented with DTP (HTP group) (*p* > 0.05) for either performance or physical characteristics of the eggs before and after storage. Both experimental diets (HT and HTP) led to a higher yolk color score (*p* < 0.001) both in fresh eggs and in those stored for 30 days at 4 °C in relation to the values measured in the eggs of control hens.

Egg storage caused losses in total egg, egg white and shell weight, while yolk weight increased (*p* < 0.05), and shell thickness and yolk color remained unchanged (*p* > 0.05) (Table 5).

### 3.3. Cholesterol Content and Yolk Fatty Acid Profile

The total lipids in the yolk were not affected (*p* > 0.05) by the dietary treatments (Table 6); instead, the cholesterol content in the yolk of HT and HTP eggs was lower (*p* < 0.05) than that in the control eggs (group C) (Figure 1).

The major changes in egg yolk FA composition as an effect of dietary treatments can be summarized as an increase in PUFA content (LA, ALA, EPA and DHA) and a decrease in MUFAs (C18:1*c*9 and C16:1), while SFAs did not change significantly (*p* > 0.05). The major lipid SFA in the yolk was palmitic acid (PA; C16:0), and the second most abundant was stearic acid (SA; C18:0). Egg yolk PA content decreased (*p* < 0.01) while SA increased (*p* < 0.01) in both HT and HTP eggs compared to values in C eggs. The increase in the proportion of PUFAs in HT and HTP eggs was due to both the increase in the proportion of n-6 FAs (*p* < 0.001) and the proportion of n-3 FAs (*p* < 0.001) compared to those in C eggs. The concentration of ALA (C18:3n-3) was almost 12 times higher (*p* < 0.001) in HTP eggs than in C eggs and 20.7% higher (*p* < 0.05) than in HT eggs. Furthermore, EPA (C20:5n-3) was almost four times higher and DHA (C22:6n-3) two times higher in HT and HTP egg yolk compared to the concentrations determined in egg yolks C (Figure 1).

No differences (*p* > 0.05) were observed between the DT-supplemented diet (HT group) and the DTP-supplemented diet (HTP group) regarding the yolk FA profile, except for n-3 FAs, which were higher in HTP eggs due to the higher content of ALA compared to that in HT eggs.

Egg storage at 4 °C for 30 days decreased the PUFA concentration and increased the MUFA concentration in egg yolks laid by hens of all groups, while SFA levels did not change during storage. Among PUFAs, the most affected were n-3 FAs, especially long-chain n-3 FAs (EPA + DHA). Compared to that in fresh eggs, the concentration of EPA + DHA in the yolk of stored eggs decreased (*p* < 0.05) by 33.3%, by 22.4% and by 15.2% in C, HT and HTP eggs, respectively. Similar decreases were also found for ALA concentration. Notably, the total and individual PUFA concentration decreases in the yolk were lower (*p* < 0.05) in the DTP-supplemented group than in the DT-fed group (Table 7).

### 3.4. Yolk Lipid Quality Indices

The qualitative, nutritional and metabolic indices of egg yolk lipids, estimated based on the fatty acid profile, were significantly (*p* < 0.001) affected by dietary treatments, except for the SI (saturation index), NVI (nutritional value index), AI (atherogenicity index) and HPI (health-promoting index) (Table 8). The qualitative and nutritional indices of lipids in the egg yolks of the HT and HTP groups were considerably higher than those of the eggs of the C group. The metabolic indices, except for the EI (elongase index), in the egg yolks of the HT and HTP hens were lower (*p* < 0.001) than those in control eggs (Table 8).

The n-6/n-3 FA ratios in egg yolks laid by hens fed the experimental diets were 3.92 (HT) and 3.43 (HTP); these values are almost three times lower than those in group C. The same phenomenon was found in the case of the LA/ALA ratio, which was 8.4 times lower in the HT group and 10.4 times lower in the HTP group than in the C group (Table 8).

The AI was not significantly affected by dietary treatments, while the TI (Thrombogenicity index) decreased significantly (*p* < 0.001) from 0.96 (C group) to 0.76 (HT group) and 0.78 (HTP group). As expected, the peroxidability index (PI) was significantly higher (*p* < 0.001) for HT and HTP egg yolk lipids than for those in group C. The ratio of hypocholesterolemic to hypercholesterolemic FAs (h/H FAs), oxidative susceptibility (OS), polyunsaturation index (PI), egg lipid quality index (LQI) and desirable FAs (DFAs) showed higher values (*p* < 0.05) in yolk samples from HT and HTP eggs than in those from C eggs.

### 3.5. Antioxidant Profile and Lipid Oxidative Status of Yolks

Table 9 shows the effects of food byproducts (HSC, DT and DTP) added to the diet of laying hens on the antioxidant profile of the yolks and the oxidative stability of the yolk lipids before and after storing the eggs for 30 days at 4 °C.

Egg yolk antioxidant concentrations (lycopene, β-carotene, lutein, retinol, total phenols and α-tocopherol) and ABTS radical-scavenging activity increased for both experimental diets (HT and HTP), while MDA content decreased compared to that in the eggs provided by hens in the control group (C). Egg yolk lycopene, β-carotene and α-tocopherol concentrations were higher in eggs laid by hens supplemented with DTP compared to those in eggs from hens fed DT. Lycopene, the main carotenoid in tomatoes, was 67.6% higher (*p* < 0.001) in the yolk of HTP eggs compared to that in HT eggs.

After 30 days of egg storage at 4 °C, the lycopene, lutein and β-carotene contents of the yolks in all types of eggs (C, HT and HTP) did not change significantly, while the content of retinol, α-tocopherol and total phenolics decreased (*p* < 0.05), which caused a significant decrease (*p* < 0.001) in the antioxidant capacity and an increase (*p* < 0.01) in the MDA content of the egg yolks. However, the supplementation of diets with tomato waste reduced the intensity of lipid peroxidation processes in the yolk during storage, so the MDA concentration was lower in HT and HTP eggs than in C eggs (Table 9) despite the fact that eggs enriched with n-3 FAs (HT and HTP eggs) are much more susceptible to lipid oxidation. Significant differences (*p* < 0.05) were found between HT and HTP eggs in favor of HTP eggs, which means that the lowest MDA content and the highest antioxidant capacity were found in eggs laid by hens supplemented with dried tomato pomace (HTP group).

### 3.6. Correlation and Principal Component Analysis (PCA)

Correlation analysis was performed to determine the relationships between cholesterol content, FA profile, antioxidant content, color and antioxidant status of egg yolks. Pearson’s correlation showed that the n-3 FA concentration in the yolk was negatively correlated with cholesterol and positively with ALA and α-tocopherol, and α-tocopherol was positively correlated with ALA. Figure 2 reveals a strong positive correlation between ABTS radical-scavenging activity and the concentrations of lycopene, phenols and α-tocopherol, while the content of yolk in MDA is correlated negatively with the concentration of antioxidants. Positive correlations were found between yolk color scores (RYCF) and lycopene and lutein concentrations (Figure 2).

We used PCA (principal component analysis) based on the correlation matrix to differentiate the variables in a multivariate data table and to classify the tested treatments. It was revealed that the first six principal components cumulatively explained 92.20% of the total variance. In the PCA results, the first two principal components explain 78.86% of the data on the FA profile, cholesterol content, yolk color, antioxidant content and antioxidant status of egg yolks. PC1 accounted for 73.16% of the variation and was positively correlated with PUFAs, n-3 FAs, n-6 FAs and ALA and negatively with MUFAs and oleic acid (Figure 3). PC2 represented 5.70% of the variation and was positively correlated with cholesterol and negatively correlated with linoleic acid. The plan projection determined via the first two PCs shows a clear discrimination between the groups of hens based on the composition of FA, color and antioxidant status of the yolks (Figure 3). The hens in the control group are located on the left side, differentiated from the experimental groups HT and HTP, which are located on the right side of the figure. The clear discrimination of the hens in the three groups under study shows a significant difference that is definitely related to the impact of incorporating hempseed cake and tomato waste in the diet.

## 4. Discussion

### 4.1. Chemical Composition of Hempseed Cake and Tomato Waste

Food waste is a major problem with important economic, social and environmental implications. Therefore, waste from food overproduction and food processing should be not only reduced but also “recycled” to ensure the sustainable use of natural resources. Consequently, the scientific community is encouraged to investigate different strategies for the management and sustainable utilization of these wastes. The present study demonstrated that the tested food wastes (hempseed cake and tomato waste) are still rich sources of nutrients and numerous phytochemicals and can be effectively used in poultry feed. However, the high fiber content of hempseed cake and tomato waste acts as an anti-nutritional factor, reducing the digestion and absorption of nutrients from feed [29,30]. According to our findings, the crude fiber content was 28.41% (% of DM) in HSC, 25.97% in DT and 38.92% in DTP, respectively, similar to the results previously reported in other studies [6,16,31,32].

DTP had a higher fat content than DT (11.83 vs. 5.12 g/100 g DM) due to the high proportion of seeds (33% of DTP composition; [33]), which have a high content of vegetable oil (up to 35%) and in which linoleic, linolenic and oleic acids dominate [34]. In addition, DTP provided larger amounts (*p* < 0.05) of bioactive compounds (lycopene, β-carotene, total phenols and α-tocopherol) and a higher ABTS radical-scavenging activity (*p* < 0.05) compared to DT, similar to the findings of Farinon et al. [35], who compared tomato pomace with whole tomato powder available on the market. These differences can be attributed to the fact that the highest concentration of bioactive compounds is in the peel (skin fraction) of tomatoes, which, according to Lu et al. [33] represents 67% of the composition of dried tomato pomace and only 14% of the composition of whole dried tomatoes.

Lycopene is a lipophilic bioactive compound responsible for the red color of vegetables (tomatoes, peppers) and is considered one of the most powerful natural antioxidants [11]. In the present study, the lycopene content was higher in DTP than in DT (945.4 mg/kg DM and 467.8 mg/kg DM, respectively) (Table 2). These results are confirmed by Sharma and Le Maguer [36], who reported that between 72% and 92% of the total lycopene content of tomatoes is in the peel (skin fraction). In addition, it has been reported that the lycopene content of DTP can vary from 10 mg/g DM to 7000 mg/g DM [37], depending on the cultivar, tomato growing practices and tomato processing conditions as well as the laboratory method used for the determination.

### 4.2. Performance of Laying Hens and Egg Quality

The results of a meta-analysis by Handayani et al. [10] showed that the incorporation of tomato pomace in the diet of laying hens up to a level of 7.1% (% of DM) significantly increased feed intake but did not change egg production or egg mass, similar to the results obtained in our study. The increase in feed intake in hens of the HT and HTP groups could be related to the presence of ascorbic acid in dried tomato waste, which increases appetite and therefore feed intake [38]. According to Colombino et al. [39], the increase in feed intake could be related to the high fiber content of DTP and HSC, which accelerated the speed of feed transit in the intestine, reducing the digestion and absorption of nutrients. This could explain the lack of significant differences compared to the control group regarding egg production, egg weight and FCR (Table 4).

The results obtained in this study showed that there was no effect of dietary treatment on egg constituent weights (albumen, yolk or shell) or shell thickness, similar to the results of Tufarelli et al. [13]. Some studies reported that the incorporation of HSC in the diet of laying hens (up to 20%) had no negative effects on egg quality traits [6,40], while other studies reported that HSC (up to 15%) decreased the yolk percentage and increased the egg white percentage [41].

Contrary to the present study, Mansoori et al. [42] reported increases in egg weight, shell weight and shell thickness in laying hens fed diets containing dried tomato pomace at a level of 10%. Furthermore, Calislar and Uygur [18] reported that tomato pomace introduced into hen feed at a level of 20% did not influence egg production or egg weight. In contrast, Panaite et al. [19] reported that the simultaneous supplementation of the diet of laying hens with linseed and dried tomato waste at a level of 5% significantly reduced daily feed intake and egg laying percentage, a conclusion that was not confirmed in the present study. Differences between the present study and other experiments regarding the effect of introducing tomato waste into the diet of laying hens on egg performance and quality may be due to several factors such as (1) tomato variety, soil type, tomato cultivation practices, stage baking and climate; (2) different methods of processing tomatoes, e.g., for the production of tomato paste, sauces, ketchup or tomato juice; (3) the rate of supplementation of laying hen diets with tomato waste; and (4) other factors (composition of diets, age of hens, genetic factors, duration of experiment).

Egg weight loss during storage occurs regardless of the storage conditions [43], which was also confirmed in our research. This was due to the biophysical and chemical changes taking place in the egg content from the moment when the eggs are laid. During storage, solvents (water and other gaseous products) from the egg are lost through evaporation, and the rate of evaporation is influenced by storage time, temperature, relative humidity and the porosity of the shell. During storage, the pH of the egg albumen increases through the release of carbon dioxide as a result of carbonic acid dissociation, which in turn leads to the weakening of bonds within the ovomucin–lysosome complex. This promotes egg weight loss [43]. The albumin weight loss during storage was due to the passage of water and gases from the albumen via the shell pores and diffusion into the yolk [44]. The weight of the yolk increased (*p* < 0.05) during storage in all types of eggs (C, HT and HTP), which could be due to the diffusion of water from the white to the yolk induced by the difference in osmotic pressure [44].

Even though the color of the egg yolk has no impact on the nutritional value of the egg, it is important for consumers because a more intense color of the yolk is associated with better nutritional value and also with a better health status of the hens [18].

The high content of carotenoid pigments, especially lycopene, in tomato waste (Table 2) was most likely responsible for the significant increase in color score (RYCF) in HT and HTP eggs compared to that in the control. These results are in agreement with those reported by Habanabashaka et al. [16], who found an increase in egg yolk color from 4.66 (control) to 9.15 after supplementing the diet of laying hens with 6% dried tomato pomace. Similar results were also obtained by Salajedheh et al. [45], who found that dietary supplementation with 150 and 190 g/kg dried tomato powder increased the egg yolk color score from 7.25 (control) to 8.5 and 9.83, respectively. In the same sense, Amar et al. [46] found that the dietary incorporation of 130 g dried tomato peel/kg DM significantly increased egg yolk color scores from 7.25 to 9.38.

Regarding the effect of storage on egg yolk color, the results obtained in the present study are in agreement with those reported by Omri et al. [47], who found that the color scores of egg yolk enriched in pigments (lycopene, zeaxanthin and carotene) did not change during egg storage for one month at 4 °C. In addition, Barbarosa et al. [48] reported that the storage of eggs enriched in n-3 FAs did not affect the yolk color of eggs stored for 35 days at 7.9 °C, but they found a significant decrease in yolk color intensity in eggs stored at room temperature (26.5 °C) starting from the 28th day of storage.

### 4.3. Egg Yolk Cholesterol Content

In the present study, the decrease in cholesterol content in the yolks of HT and HTP eggs could be explained by the fact that hempseed cakes contain phytosterols, especially sitosterol, which limits cholesterol absorption [49,50,51]. Furthermore, Golimowski et al. [52] and Vlaicu et al. [53] concluded that phytosterols in the diet of laying hens reduce cholesterol biosynthesis in the liver of hens, limiting the amount of cholesterol in the yolk. The reduction in yolk cholesterol concentration in the present study may also be due to the increased intake of phenolic compounds present in DT or DTP, which may reduce endogenous cholesterol synthesis [54]. Habanabashaka et al. [16] reported a decrease in yolk cholesterol concentration when hens’ diets were supplemented with DTP, probably due to the high dietary fiber content. Numerous other studies claim that a decrease in cholesterol content in yolks is correlated with an increased concentration of n-3 FAs in hen feed [6,55,56,57].

The results of the present study are in agreement with those found by Mierlita et al. [4], who reported that the inclusion of hemp seeds in the diet of hens at a level of 8% decreased the cholesterol concentration in the yolk. Similar results were also reported by Basmacioglu et al. [55], who found that the inclusion of flaxseed in the diet of hens at a level of 8.64% decreased the cholesterol concentration in the yolk but had no effect at a level of inclusion of 4.32%.

### 4.4. Egg Yolk Fatty Acid Profile

The major SFA of yolk lipids was PA (palmitic acid; C16:0), and the second most abundant was SA (stearic acid; C18:0). Egg yolk PA content decreased (*p* < 0.01) while that of SA increased (*p* < 0.01) in both HT and HTP eggs compared to levels in C eggs. It follows that palmitic acid (C16:0) synthesized at the tissue level was esterified and converted to stearic acid (C18:0) to a greater extent in HT and HTP hens than in C hens [58]. Similar to the results obtained in the present study, a reduction in the concentration of PA and an increase in the level of SA in the egg yolk was previously reported [6] in response to the incorporation of hemp traces in the diet of laying hens.

High concentrations of n-3 FAs in hen diets can inhibit the activity of the enzyme (stearoyl-CoA desaturase) that ensures the conversion of C16:0 to C16:1 and of C18:0 to C18:1 [3], thus causing, in our study, a decrease in the concentration of the two FAs (palmitoleic and oleic acids) and of total MUFAs in the yolks of eggs laid by hens that received HSC in their diet (HT and HTP groups). The decrease in OA concentration in the yolk of HT and HTP eggs found in the present study is in agreement with the research of Yalcin [59], who reported that supplementing the diet of laying hens with n-3 FAs reduced the OA content of the yolk by inhibiting the enzyme activity of ∆9-desaturases.

The concentration of n-3 FAs in the yolk of fresh eggs was higher (*p* < 0.001) in the experimental groups than in the control group: 3.38 times higher for HT and 3.76 times higher for HTP. In addition, the n-6/n-3 ratio decreased from 11.23 (C) to 3.92 (HT) and 3.43 (HTP). These results are similar to those reported by Panaite et al. [19] when the diet of laying hens was supplemented with linseed and different levels of dried tomato waste.

The concentrations of n-6 FAs and n-3 FAs in the yolk are influenced by the fatty acids provided by the hens’ diet but also by the availability of enzymes responsible for the desaturation and elongation of fatty acids in the liver of hens [60]. In the metabolism of fatty acids, there is competition for desaturase enzymes. For example, ALA competes with LA for the same ∆6-desaturase enzyme that converts ALA to EPA and LA to AA (arachidonic acid, C20:4n-6) [61]. In our study, increasing the content of ALA in the experimental diets (HT and HTP) led to an increase (*p* < 0.001) in the concentration of EPA in the yolk, probably due to an increase in the activity of the desaturase enzyme used in the conversion of ALA to EPA [57,62]. The experimental diets (HT and HTP) also caused a decrease in the AA content of the yolk compared to that of the control hens, probably due to the decrease in the availability of the enzyme ∆6-desaturase, which reduced the conversion rate of LA in AA. The conversion of ALA to EPA and DHA is favored by a low n-6/n-3 FA ratio; otherwise, the LA to AA conversion pathway is promoted [5]. Our study confirmed this conclusion; the low n-6/n-3 FA ratio favored the conversion of ALA to EPA and DHA in the case of the experimental groups (HT and HTP) (Table 6 and Table 7) compared to the control group, in which a greater amount of AA was observed in egg yolks.

The increase in the yolk GLA (gamma-linolenic acid, C18:3 n-6) concentration found in this study for hens fed hempseed cake was also reported by Taaifi et al. [5] and is explained by the larger amount of GLA present in hemp seeds compared to that of other oilseeds [6].

The results obtained in this study are in general agreement with the reports of previous studies, which found that storing n-3 FA-enriched eggs in the refrigerator for 20–60 days reduces the concentration of PUFAs and increases that of MUFAs in the yolk [63,64]. Changes observed in the FA profile of eggs stored for 30 days at 4 °C indicate that the antioxidants in tomato waste (DT and DTP) slowed down the oxidation process of egg yolk fat. In addition, DTP was found to be more effective than DT in preventing yolk lipid peroxidation due to its better protection of FAs valuable to human health, such as total and individual n-3 FAs. The results of our study are supported by the research of Toyes-Vargas et al. [65], who found that seaweed is more effective in improving the n-3 FA content of egg yolk compared to fishmeal due to the carotenoid content in seaweeds that reduce n-3 FA oxidation in the yolk during refrigerated egg storage.

### 4.5. Egg Yolk Lipid Quality Indices

It is desirable that the PUFA/SFA ratio in egg yolk be high on the assumption that PUFAs lower LDL and total cholesterol and SFAs can increase serum cholesterol in humans. In this sense, eggs laid by hens fed the experimental diets (HT and HTP) provide a greater benefit to cardiovascular health compared to control eggs. In the current study, dietary treatments (HT and HTP) decreased the ratio of n-6/n-3 FAs in egg yolks from 11.23 (C group) to 3.92 (HT group) and 3.43 (HTP group) (*p* < 0.001), so that eggs laid by hens fed the experimental diets (HT and HTP) fell within the n-6/n-3 FA ratio of 4:1 recommended for human health [1].

In the present study, egg yolks from HT and HTP hens had a higher UI (unsaturation index) value than those from control hens, indicating a higher probability of FA autoxidation. On the other hand, the oxidation susceptibility index (PI) values found in this study were significantly higher in eggs laid by hens fed diets enriched in PUFAs and antioxidants (HT and HTP) compared to those in C eggs, which indicates a high predisposition of yolk lipids to oxidation and shortened shelf life.

Eggs with desirable human health benefits are defined by lower AI and TI values and higher h/H FA (hypocholesterolemic/hypercholesterolemic FA) values [66].

In the present study, a significant decrease in the TI and an increase in the h/H FA ratio could indicate an improvement in the nutritional quality of egg yolks. Similar values for AI, TI and h/H FAs were previously reported by Mierlita et al. [4] in hens fed 8% hemp seeds and 3% dried fruit pomace (rose hips and black currants).

The experimental diets did not affect the AI value, probably due to the increase in the concentration of stearic acid in the yolks of HT and HTP eggs, which is not considered proatherogenic due to the ability of humans to desaturate stearic acid to oleic acid [1]. On the other hand, the HT and HTP experimental groups had lower levels of Δ9-desaturase, which converts saturated FAs (C16:0+C:18:0) to monounsaturated FAs (C16:1+C:18:1), suggesting that the lower OA and PA levels in HT and HTP egg yolks could be due to decreased Δ9-desaturase activity [66].

### 4.6. Lipid Oxidative Status of Yolks

In agreement with previous studies by Panaite et al. [19] and Mierlita et al. [4], the results obtained in the present study showed that the simultaneous enrichment of the diet of laying hens with n-3 FAs and natural antioxidants (carotenoids, tocopherols and phenols) leads to eggs with a higher content of n-3 FAs and antioxidants, improved antioxidant activity and better PUFA stability during egg storage.

Although increasing the level of n-3 FAs in the yolk is a key goal in achieving healthier eggs, the high content of unsaturated FAs in the yolk could lead to increased susceptibility to lipid oxidation during storage [4,13]. In the present study, even though the experimental HT and HTP diets led to an increase in the level of n-3 FAs (ALA, EPA and DHA), the concentration of MDA (product resulting from lipid oxidation) in the yolk decreased significantly. These results could be attributed to the bioactive compounds present both in HSC (such as tocopherols, phenols and cannabidiol) [67] and tomato waste (such as lycopene, lutein, tocopherols and phenols) that exert powerful antioxidant activities [13].

Similar to the results obtained in this study, Habanabashaka et al. [16] reported that tomato pomace included in the diet of hens at a level of 6% had no adverse effect on egg production but increased the concentration of lycopene, lutein and β-carotene in egg yolks and significantly reduced the concentration of MDA.

The dietary addition of hempseed cake and tomato waste increased (*p* < 0.001) the antioxidant activity of the egg yolks as well as the oxidative stability of the lipids in the yolks. Thus, the enrichment of eggs with n-3 FAs did not reduce the oxidative stability of lipids in the egg yolks due to the higher concentrations of antioxidants (lycopene, lutein, β-carotene, retinol, α-tocopherol and total phenols) in the yolks as a result of supplementing the diet with tomato waste (DT and DTP). It is noteworthy that DTP was more effective than DT in reducing the MDA content in the yolks of fresh and stored eggs. This superior effect of DTP could be related to its significantly higher content of lycopene, β-carotene, phenols and α-tocopherol (Table 2), which were efficiently transferred to the yolks. Similarly, a study carried out by Bianchi et al. [68] demonstrated that tomato skin and seeds (tomato pomace) are significantly richer in antioxidants and have higher antioxidant activity than tomato pulp. In addition, a study by Szabo et al. [69] reported that in tomato peel, the concentration of lycopene and β-carotene is five times higher than that found in the pulp fraction. Lycopene has a singlet oxygen-quenching capacity 10 times higher than that of α-tocopherol and 2 times higher than that of β-carotene [70], being one of the most effective antioxidants.

Omri et al. [47] evaluated the effect of the dietary incorporation of flaxseed together with tomato and pepper paste powder mixture and reported that the storage of eggs for 30 days at 4 °C did not affect (*p* > 0.05) carotenoid concentrations (lycopene, β-carotene and zeaxanthin) in egg yolks but decreased (*p* < 0.05) the concentrations of phenols and α-tocopherol, which is in agreement with the results of our study. It has been suggested that the decrease in α-tocopherol concentration may be related to the involvement of α-tocopherol in protecting PUFAs against oxidation, reducing the amount of this antioxidant available in the egg yolks [71].

In our study, the significant decrease (*p* < 0.05) in the concentration of unsaturated FAs and mainly polyunsaturated FAs (n-6 FAs, n-3 FAs, ALA, EPA + DHA) in the yolks and the increase in the level of MDA indicated the occurrence of lipid oxidation during storage for all types of eggs. We can deduce that the antioxidants from tomato waste (DT and DTP) did not prevent the oxidation processes of yolk lipids but reduced their intensity. Thus, the stored eggs obtained from hens in the experimental groups (HT and HTP) showed a smaller increase in the concentration of MDA in the yolks (+42.6% and +39.3%, respectively) than that of the control group (+62.4%), even though the concentration of PUFAs in the yolks was significantly higher in the experimental groups (Table 9). The antioxidants present in tomato waste act as free-radical scavengers, limiting peroxidation in egg yolk lipids and the production of MDA. Antioxidants can donate a hydrogen atom, causing the direct quenching of ROS (reactive oxygen species) before they can damage fatty acids, or they can block the action of enzymes (for example, xanthine oxidase and protein kinase C) that directly generate O2 [72]. It should be emphasized that the highest oxidative stability of yolk lipids was observed in eggs laid by hens fed additionally with DTP, which also provided the highest concentration of antioxidants (in particular, lycopene, β-carotene and α-tocopherol) in the yolks of stored eggs. In line with our results, other previous studies also demonstrated that natural antioxidants from different sources (tomato waste, rosehip meal, grape pomace, fruit pomace and sea buckthorn meal) are effective in reducing yolk MDA concentrations during egg storage [4,9,12,71].

## 5. Conclusions

The obtained results indicate that tomato waste (DT and DTP) contains significant amounts of bioactive phytochemicals (carotenoids, phenols and α-tocopherol) with high antioxidant potential.

The incorporation of HSC at a level of 20% in the diet of hens as a source of PUFAs and the supplementation of the diet with 4% tomato waste caused a significant increase in Roche color score of the yolks, in correlation with increased deposition of lycopene, lutein and β-carotene in egg yolks. In addition, supplementing the diet of hens with dried tomato waste improved the antioxidant activity of the yolks, which ensures the preservation of the quality of eggs enriched with unsaturated FAs during storage for 30 days at 4 °C. DTP showed a higher antioxidant content and a higher potential to prevent yolk lipid peroxidation than DT.

Further studies are needed to determine the impact of tomato waste on digestion processes and the immune system of laying hens, as well as the stability of bioactive compounds during the cooking of eggs.

## Figures and Tables

**Figure 1 animals-14-03444-f001:**
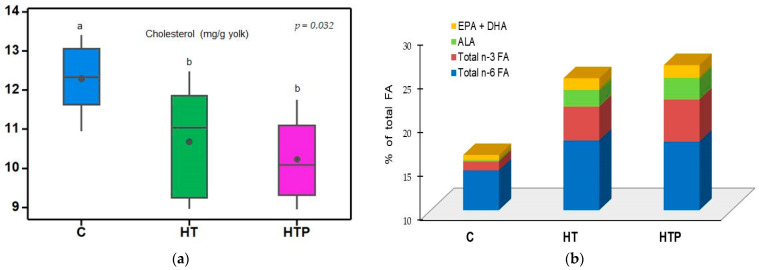
Effect of feeding treatment on yolk contents of cholesterol ((**a**); ^a,b^ means within a row with different superscripts are different (*p* < 0.05)) and major PUFAs (% of total fatty acids) (**b**).

**Figure 2 animals-14-03444-f002:**
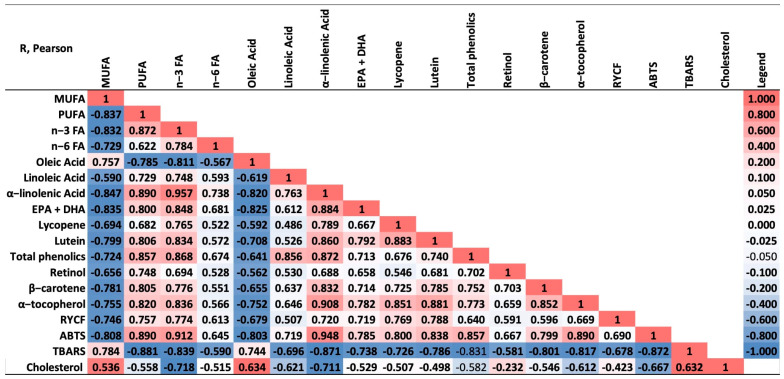
Pearson correlations of yolk egg characteristics.

**Figure 3 animals-14-03444-f003:**
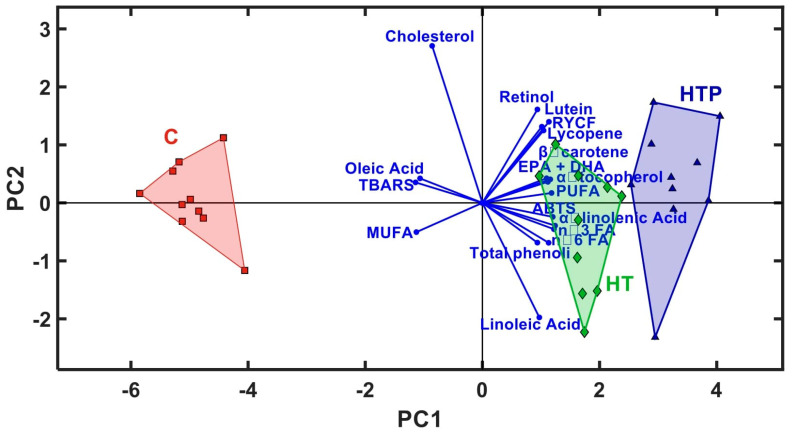
Principal component analysis (first and second discriminant function) plot for cholesterol content, FA profile, antioxidant composition, color and antioxidant status of egg yolks.

**Table 1 animals-14-03444-t001:** Ingredients and chemical composition of the diets.

Items	Experimental Diets ^1^		
	C	HT	HTP
Ingredients (% as fed)			
Maize	45.60	41.42	41.67
Wheat	10.00	10.00	10.00
Soybean meal	20.50	12.70	12.45
Sunflower meal	9.60	-	-
Sunflower oil	2.40	-	-
Hempseed cake	-	20.00	20.00
Dried whole tomato (DT)	-	4.00	-
Dried tomato pomace (DTP)	-	-	4.00
DL-Methionine	0.15	0.11	0.11
L-Lysine	0.10	0.12	0.12
Choline premix ^2^	0.05	0.05	0.05
Ronozyme WX ^3^	0.02	0.02	0.02
Calcium carbonate	8.90	8.90	8.90
Monocalcium phosphate	1.38	1.38	1.38
Salt	0.30	0.30	0.30
Vitamin–mineral premix ^4^	1.00	1.00	1.00
Composition (calculated unless noted) ^5^			
Metabolizable energy (kcal/kg) ^6^	2769	2752	2765
Crude protein (CP) (analyzed, %)	17.63	17.54	17.60
Ether extract (EE) (analyzed, %)	4.38	5.36	5.71
NDF (analyzed, %)	12.13	16.76	17.30
ADF (analyzed, %)	6.56	9.68	10.42
Lysine (%)	0.84	0.84	0.84
Methionine (%)	0.40	0.40	0.40
Cystine (%)	0.28	0.28	0.28
Threonine (%)	0.67	0.67	0.67
Tryptophan (%)	0.17	0.17	0.17
Calcium (%)	3.91	3.91	3.91
Total phosphorus (%)	0.65	0.65	0.65
Available phosphorus (%)	0.45	0.45	0.45
Sodium (%)	0.15	0.15	0.15

^1^ C: standard diet; HT: diet containing 20% HSC and 4% DT (dried whole tomato); HTP: diet containing 20% HSC and 4% DTP (dried tomato pomace); NDF: neutral detergent fiber, ADF: Acid detergent fiber. ^2^ Premix contained 180,000 mg/kg of choline chloride. ^3^ Ronozyme WX is an additive that contains endo-1,4-β-xylanase (EC 3.2.1.8) and ensures 100 FXU (fungal xylanase units)/kg feed. ^4^ Vitamin–mineral premix provided per kilogram of diet: vit. A 12,000 IU; vit. D3 2800 IU; vit. E 150 IU; vit. K 6 mg; vit. B12 0.04 mg; vit. B2 4.0 mg; biotin 0.3 mg; vit. B4 1.4 mg of choline; vit. B9 3 mg; vit. B5 15 mg; niacin 32 mg; Fe 60 mg; Se 0.5 mg; I 0.5 mg; Zn 80 mg; Mn 60 mg; Cu 10 mg. ^5^ Calculated based on NRC feedstuff nutrient tables [20]. ^6^ Metabolizable energy (kcal/kg) = 2707.71 + 58.63 EE − 16.06 NDF [21].

**Table 2 animals-14-03444-t002:** Chemical composition and antioxidant profile of hempseed cake (HSC) and tomato waste (dried whole tomato: DT; dried tomato pomace: DTP).

	HSC	DT	DTP	*p*-Values
Water content (g/kg fresh)	86.21 ± 1.8 ^c^	952.6 ± 4.2 ^a^	691.8 ± 3.1 ^b^	˂0.001
Proximate composition (% of DM)				
Dry matter (DM, %)	91.38 ± 1.79	90.42 ± 2.04	92.17 ± 2.75	0.357
Crude protein (CP)	32.21 ± 2.47 ^a^	17.71 ± 0.67 ^c^	20.33 ± 1.34 ^b^	˂0.001
Ether extract (EE)	14.32 ± 1.74 ^a^	5.12 ± 0.32 ^c^	11.83 ± 0.98 ^b^	˂0.001
Crude fiber	28.41 ± 1.39 ^b^	25.97 ± 1.27 ^c^	38.92 ± 1.83 ^a^	˂0.001
Crude ash	6.18 ± 0.47	5.41 ± 0.55	5.98 ± 0.27	0.084
NDF (neutral detergent fiber)	43.74 ± 1.67 ^b^	45.62 ± 1.98 ^b^	58.43 ± 2.09 ^a^	˂0.001
ADF (acid detergent fiber)	30.26 ± 0.54 ^b^	26.18 ± 1.12 ^c^	44.59 ± 0.98 ^a^	˂0.001
Starch	10.30 ± 0.75 ^a^	4.22 ± 0.17 ^b^	1.97 ± 0.11 ^c^	˂0.001
Sugar	1.72 ± 0.17 ^c^	31.72 ± 0.61 ^a^	22.11 ± 0.70 ^b^	˂0.001
Gross energy (GE) (kcal/kg DM) *	5344	4649	5103	-
AMEn (kcal/kg DM) **	2830	2230	2489	-
Antioxidants profile (on dry matter basis)				
Lycopene (mg/kg)	ND	467.8 ^b^	945.4 ^a^	˂0.001
Lutein (mg/kg)	2.38 ^b^	12.72 ± 1.04 ^a^	14.53 ± 2.15 ^a^	˂0.001
β-Carotene (mg/kg)	6.81 ^c^	70.3 ± 0.87 ^b^	122.4 ± 7.63 ^a^	˂0.001
Total carotenoid content (mg/kg)	9.19 ^c^	550.82 ± 13.81 ^b^	1082.33 ± 57.1 ^a^	˂0.001
Total phenols (mg GAE/100 g)	58.71 ± 0.55 ^c^	127.4 ± 4.06 ^b^	174.1 ± 6.41 ^a^	˂0.001
α-Tocopherol (mg/kg)	37.92 ± 0.18 ^c^	82.34 ± 0.85 ^b^	175.27 ± 3.73 ^a^	˂0.001
ABTS radical-scavenging activity (µmol TE/100 g)	99.52 ± 2.30 ^c^	144.58 ± 4.58 ^b^	217.08 ± 6.68 ^a^	˂0.001

* Calculated values according to NRC [20]; ** Calculated values according to the equation proposed by the World’s Poultry Science Association [24]; AMEn: apparent metabolizable energy corrected for nitrogen balance; GAE: gallic acid equivalent; TE: Trolox equivalent; ND: not detectable. ^a–c^ Means within a row for the same type of feed with different superscripts are different (*p* < 0.05).

**Table 3 animals-14-03444-t003:** Fatty acid (FA) profile (% of total FA) of hempseed cake (HSC), tomato waste (dried whole tomato: DT; dried tomato pomace: DTP) and experimental diets.

	HSC	DT	DTP	SEM ^2^	*p*-Value	Experimental Diets ^1^			SEM ^2^	*p*-Value
						C	HT	HTP		
C16:0	6.48 ^b^	16.78 ^a^	15.21 ^a^	0.583	˂0.001	14.35	10.79	10.03	0.354	0.201
C18:0	2.61 ^b^	5.18 ^a^	5.66 ^a^	0.123	0.028	3.04	2.80	3.12	0.126	0.164
C18:1 *cis*-9	9.78 ^b^	18.07 ^a^	19.34 ^a^	0.914	0.009	22.41 ^a^	18.41 ^b^	15.64 ^c^	0.323	˂0.001
C18:2 n-6	56.38 ^a^	50.72 ^b^	52.62 ^b^	1.816	0.042	52.30 ^a^	47.68 ^c^	49.81 ^b^	0.838	0.003
C18:3 n-3	20.91	3.21	2.98	1.173	˂0.001	3.68	15.24	17.62	0.442	˂0.001
SFAs	10.25 ^b^	22.29 ^a^	21.79 ^a^	0.941	˂0.001	18.07 ^a^	14.40 ^b^	14.60 ^b^	0.263	0.007
MUFAs	10.85 ^b^	19.27 ^a^	20.54 ^a^	0.875	˂0.001	24.69 ^a^	20.72 ^b^	16.18 ^c^	0.197	˂0.001
PUFAs	78.66 ^a^	56.14 ^b^	57.35 ^b^	1.734	˂0.001	56.71 ^c^	64.15 ^b^	69.09 ^a^	1.072	˂0.001
n-6 FAs	57.68 ^a^	52.64 ^b^	53.58 ^b^	1.801	0.036	52.79 ^a^	48.62 ^b^	50.98 ^b^	0.954	0.005
n-3 FAs	20.98 ^a^	3.50 ^b^	3.77 ^b^	0.257	˂0.001	3.92 ^c^	15.53 ^b^	18.11 ^a^	0.313	˂0.001
n-6/n-3 FAs	2.75 ^b^	15.04 ^a^	14.21 ^a^	0.054	˂0.001	13.46 ^a^	3.13 ^b^	2.81 ^b^	0.238	˂0.001

^1^ C: standard diet; HT: diet containing 20% HSC and 4% DT; HTP: diet containing 20% HSC and 4% DTP. ^2^ SEM: standard error of the mean. ^a–c^ Means within a row for the same type of feed with different superscripts are different (*p* < 0.05).

**Table 4 animals-14-03444-t004:** Effects of hempseed cake and tomato waste (dried whole tomato and dried tomato pomace) on laying hen performance (average values/group).

	Dietary Treatment ^1^			SEM ^2^	*p*-Values
	C	HT	HTP		
Body weight change (g/70 d)	+116.4	+123.1	+97.5	5.152	0.487
Daily feed intake (g/day/hen)	114.18 ^b^	117.51 ^a^	116.67 ^a^	1.385	0.037
FCR (kg feed/kg egg)	2.117	2.151	2.148	0.032	0.078
Daily egg production (%)	90.70	90.42	90.91	0.541	0.159
Average egg weight (g)	59.45	60.40	59.72	0.178	0.681
Daily egg mass (g/day/hen)	53.92	54.61	54.29	0.204	0.352

^1^ C: standard diet; HT: diet containing 20% HSC and 4% DT (dried whole tomato); HTP: diet containing 20% HSC and 4% DTP (dried tomato pomace). ^2^ SEM: standard error of the mean; FCR: feed conversion ratio. ^a,b^ Means within a row with different superscripts are different (*p* < 0.05).

**Table 5 animals-14-03444-t005:** Effects of hempseed cake and tomato waste (dried whole tomato and dried tomato pomace) on the physical characteristics of eggs stored for 30 days at 4 °C.

Parameters		Dietary Treatment ^1^			SEM ^2^	*p*-Value ^3^
		C	HT	HTP		
Average egg weight (g)	Fresh	59.45	60.40	59.72	0.178	0.768
	Stored	57.67	58.22	57.75	0.224	0.137
	SEM ^2^	0.201	0.098	0.136	-	-
	*p*-value ^4^	0.037	0.024	0.041	-	-
Albumen weight (g)	Fresh	38.29	38.66	38.02	0.398	0.237
	Stored	35.23	35.95	35.86	0.152	0.365
	SEM ^2^	0.428	0.352	0.391	-	-
	*p*-value ^4^	0.028	0.043	0.031	-	-
Yolk weight (g)	Fresh	14.32	14.53	14.65	0.081	0.494
	Stored	15.63	15.72	15.38	0.053	0.179
	SEM ^2^	0.063	0.047	0.022	-	-
	*p*-value ^4^	0.008	0.005	0.027	-	-
Shell weight (g)	Fresh	6.84	7.22	7.05	0.042	0.098
	Stored	6.31	6.55	6.51	0.070	0.244
	SEM ^2^	0.041	0.011	0.026	-	-
	*p*-value ^4^	0.007	0.005	0.008	-	-
Shell thickness (mm)	Fresh	0.362	0.371	0.364	0.001	0.084
	Stored	0.356	0.359	0.348	0.001	0.176
	SEM ^2^	0.001	0.005	0.008	-	-
	*p*-value ^4^	0.078	0.152	0.067	-	-
Roche yolk color fan score (RYCF)	Fresh	6.67 ^b^	8.74 ^a^	9.26 ^a^	0.181	˂0.001
	Stored	6.31 ^b^	8.62 ^a^	8.88 ^a^	0.214	˂0.001
	SEM ^2^	0.121	0.311	0.274	-	-
	*p*-value ^4^	0.093	0.324	0.121	-	-

^1^ C: standard diet; HT: diet containing 20% HSC and 4% DT (dried whole tomato); HTP: diet containing 20% HSC and 4% DTP (dried tomato pomace). ^2^ SEM: standard error of the mean. ^3^ Effect of diets. ^4^ Effect of storage. ^a,b^ Means within a row with different superscripts are different (*p* < 0.05).

**Table 6 animals-14-03444-t006:** Effect of hempseed cake and tomato waste (dried whole tomato and dried tomato pomace) on the fatty acid (FA) profile of egg yolks.

		Dietary Treatment ^1^			SEM ^2^	*p*-Value
		C	HT	HTP		
Total fat (g/100 g)		28.41	29.01	29.42	1.298	0.614
Fatty acids (% of total FAs)						
	C12:0	0.18	0.16	0.19	0.012	0.258
	C14:0	0.26	0.28	0.28	0.027	0.406
	C15:0	0.07	0.05	0.07	0.012	0.123
	C16:0	24.27 ^a^	22.80 ^b^	22.72 ^b^	0.713	0.008
	C17:0	0.26	0.28	0.29	0.033	0.691
	C18:0	8.71 ^b^	10.41 ^a^	10.59 ^a^	0.710	0.007
Σ SFAs		35.75	33.98	34.14	0.417	0.473
	C14:1	0.03	0.03	0.02	0.001	0.200
	C16:1	3.48 ^a^	2.31 ^b^	2.17 ^b^	0.202	˂0.001
	C17:1	0.12	0.10	0.11	0.006	0.375
	C18:1 *cis*-9 (OA)	40.30 ^a^	36.51 ^b^	36.62 ^b^	0.883	˂0.001
Σ MUFAs		43.93 ^a^	38.95 ^b^	38.92 ^b^	0.562	˂0.001
	C18:2 n-6 (LA)	14.57 ^b^	17.63 ^a^	17.27 ^a^	0.936	˂0.001
	C18:3 n-6 (GLA)	0.15 ^b^	0.18 ^a^	0.18 ^a^	0.009	0.041
	C20:3 n-6	0.21	0.22	0.24	0.088	0.362
	C20:4 n-6 (AA)	1.35 ^a^	1.18 ^b^	1.20 ^b^	0.041	0.018
Σ n-6 PUFAs		16.28 ^b^	19.21 ^a^	18.71 ^a^	0.261	˂0.001
	C18:3 n-3 (ALA)	0.23 ^c^	2.36 ^b^	2.85 ^a^	0.214	˂0.001
	C20:3 n-3	0.41 ^b^	0.85 ^a^	0.90 ^a^	0.065	˂0.001
	C20:5 n-3 (EPA)	0.12 ^b^	0.51 ^a^	0.48 ^a^	0.048	˂0.001
	C22:6 n-3 (DHA)	0.69 ^b^	1.18 ^a^	1.22 ^a^	0.061	˂0.001
Σ n-3 PUFAs		1.45 ^c^	4.90 ^b^	5.45 ^a^	0.147	˂0.001
Σ PUFAs		17.73 ^b^	24.11 ^a^	24.16 ^a^	0.816	˂0.001
Other FAs		2.59	2.96	2.78	0.026	0.276
Σ Unsaturated FAs (UFAs)		61.66 ^b^	63.06 ^a^	63.08 ^a^	1.638	0.042
Hypercholesterolemic FAs ^3^		24.71 ^a^	23.24 ^b^	23.19 ^b^	0.932	0.038
Hypocholesterolemic FAs ^4^		58.03 ^b^	60.62 ^a^	60.78 ^a^	1.418	0.017

^1^ C: standard diet; HT: diet containing 20% HSC and 4% DT (dried whole tomato); HTP: diet containing 20% HSC and 4% DTP (dried tomato pomace). ^2^ SEM: standard error of the mean. OA: oleic acid; LA: linoleic acid; GLA: gamma-linolenic acid; ALA: α-linolenic acid; AA: arachidonic acid; EPA: eicosapentaenoic acid; DHA: docosahexaenoic acid. ^3^ Hypercholesterolemic FA: (C12:0 + C14:0 + C16:0). ^4^ Hypocholesterolemic FA: (C18:1 + PUFA). ^a–c^ Means within a row with different superscripts are different (*p* < 0.05).

**Table 7 animals-14-03444-t007:** Egg yolk FA composition (% of total FA) before and after storage for 30 days at 4 °C.

		Dietary Treatment ^1^			SEM ^2^	*p*-Value ^3^
		C	H	HB		
Total SFAs	Fresh	35.75	33.98	34.14	0.417	0.473
	Stored	36.51 ^a^	33.46 ^b^	33.84 ^b^	0.352	0.008
	SEM ^2^	0.287	0.315	0.520	-	-
	*p*-values ^4^	0.271	0.528	0.085	-	-
Total MUFAs	Fresh	43.93 ^a^	38.95 ^b^	38.92 ^b^	0.562	˂0.001
	Stored	45.15 ^a^	41.68 ^b^	40.14 ^b^	0.608	0.007
	SEM ^2^	0.484	0.517	0.295	-	-
	*p*-values ^4^	0.032	0.017	0.040	-	-
Total PUFAs	Fresh	17.73 ^b^	24.11 ^a^	24.16 ^a^	0.816	˂0.001
	Stored	15.59 ^b^	22.14 ^a^	23.05 ^a^	0.428	˂0.001
	SEM ^2^	0.396	0.505	0.418	-	-
	*p*-values ^4^	0.007	0.008	0.026	-	-
Total n-6 FAs	Fresh	16.28 ^b^	19.21 ^a^	18.71 ^a^	0.261	˂0.001
	Stored	14.51 ^b^	17.92 ^a^	17.79 ^a^	0.369	˂0.001
	SEM ^2^	0.290	0.425	0.187	-	-
	*p*-values ^4^	0.005	0.007	0.044	-	-
Total n-3 FAs	Fresh	1.45 ^c^	4.90 ^b^	5.45 ^a^	0.147	˂0.001
	Stored	1.04 ^c^	3.87 ^b^	4.82 ^a^	0.283	˂0.001
	SEM ^2^	0.139	0.622	0.415	-	-
	*p*-values ^4^	0.006	0.008	0.037	-	-
EPA + DHA	Fresh	0.81 ^b^	1.69 ^a^	1.70 ^a^	0.107	˂0.001
	Stored	0.54 ^c^	1.31 ^b^	1.44 ^a^	0.091	˂0.001
	SEM ^2^	0.088	0.026	0.123	-	-
	*p*-values ^4^	0.004	0.007	0.023	-	-
ALA (C18:3 n-3)	Fresh	0.23 ^c^	2.36 ^b^	2.85 ^a^	0.214	˂0.001
	Stored	0.19 ^c^	1.92 ^b^	2.47 ^a^	0.149	˂0.001
	SEM ^2^	0.012	0.028	0.018	-	-
	*p*-values ^4^	0.018	0.035	0.028	-	-
n-6/n-3 FAs	Fresh	11.23 ^a^	3.92 ^b^	3.43 ^c^	1.407	˂0.001
	Stored	13.95 ^a^	4.63 ^b^	3.69 ^c^	1.182	˂0.001
	SEM ^2^	1.072	0.072	0.188	-	-
	*p*-values ^4^	0.033	0.271	0.094	-	-

^1^ C: standard diet; HT: diet containing 20% HSC and 4% DT (dried whole tomato); HTP: diet containing 20% HSC and 4% DTP (dried tomato pomace). ^2^ SEM: standard error of the mean. ^3^ Effect of diets. ^4^ Effect of storage; FA: fatty acid; SFA: saturated FA; MUFA: monounsaturated FA; PUFA: polyunsaturated FA; EPA: eicosapentaenoic acid; DHA: docosahexaenoic acid; ALA: α-linolenic acid. ^a–c^ Means within a row with different superscripts are different (*p* < 0.05).

**Table 8 animals-14-03444-t008:** Effect of hempseed cake and tomato waste (dried whole tomato and dried tomato pomace) on lipid qualitative, nutritional and metabolic indexing in the yolks of eggs.

	Dietary Treatment ^1^			SEM ^2^	*p*-Value
	C	HT	HTP		
Qualitative indices					
PUFAs/SFAs	0.496 ^b^	0.709 ^a^	0.708 ^a^	0.027	˂0.001
n-6/n-3 FAs	11.23 ^a^	3.92 ^b^	3.43 ^c^	1.407	˂0.001
LA/ALA	63.35 ^a^	7.47 ^b^	6.06 ^c^	1.082	˂0.001
Saturation index (SI)	0.539	0.531	0.532	0.316	0.208
Unsaturation index (UI)	15.03 ^b^	22.35 ^a^	22.97 ^a^	1.274	˂0.001
Oxidative susceptibility (OS)	722.58 ^b^	1068.30 ^a^	1101.07 ^a^	12.351	˂0.001
Peroxidability index (PI)	29.31 ^b^	43.04 ^a^	44.00 ^a^	2.296	˂0.001
Nutritional indices					
Nutritional value index (NVI)	2.019	2.057	2.078	0.096	0.371
Atherogenicity index (AI)	0.551	0.544	0.546	0.036	0.187
Thrombogenicity index (TI)	0.962 ^a^	0.760 ^b^	0.780 ^b^	0.047	˂0.001
Egg lipid quality index (LQI)	2.099 ^b^	4.209 ^a^	4.242 ^a^	0.123	˂0.001
h/H FAs	2.348 ^b^	2.608 ^a^	2.621 ^a^	0.042	0.011
Health-promoting index (HPI)	2.419	2.619	2.303	0.094	0.072
Desirable FAs (DFAs)	70.37 ^b^	73.47 ^a^	73.67 ^a^	0.355	0.006
Metabolic indices					
Elongase index (EI)	35.88 ^b^	45.66 ^a^	46.61 ^a^	0.723	˂0.001
Thioesterase index (ThI)	9335 ^a^	8143 ^b^	8114 ^b^	74.618	0.004
∆9-Desaturase (C16:1+C18:1)	57.03 ^a^	53.89 ^b^	46.20 ^c^	0.512	˂0.001
Activity index	6.30 ^a^	2.08 ^b^	1.91 ^b^	0.088	˂0.001

^1^ C: standard diet; HT: diet containing 20% HSC and 4% DT (dried whole tomato); HTP: diet containing 20% HSC and 4% DTP (dried tomato pomace). ^2^ SEM: standard error of the mean. FA: fatty acid; SFA: saturated FA; PUFA: polyunsaturated FA; LA: linoleic acid; ALA: α-linolenic acid; h/H FA: hypocholesterolemic/Hypercholesterolemic ratio. ^a–c^ Means within a row with different superscripts are different (*p* < 0.05).

**Table 9 animals-14-03444-t009:** Egg yolk antioxidant profile and lipid oxidative status before and after storage for 30 days at 4 °C.

		Dietary Treatment ^1^			SEM ^2^	*p*-Value ^3^
		C	HT	HTP		
Antioxidants profile of the yolk						
Lycopene (µg/g)	Fresh	3.89 ^c^	5.54 ^b^	9.32 ^a^	0.081	˂0.001
	Stored	3.47 ^c^	4.93 ^b^	8.95 ^a^	0.048	˂0.001
	SEM ^2^	0.035	0.041	0.020	-	-
	*p*-value ^4^	0.228	0.354	0.203	-	-
Lutein (µg/g)	Fresh	2.86 ^c^	4.74 ^b^	6.12 ^a^	0.062	˂0.001
	Stored	2.40 ^c^	4.39 ^b^	6.17 ^a^	0.059	˂0.001
	SEM ^2^	0.023	0.047	0.022	-	-
	*p*-value ^4^	0.104	0.057	0.201	-	-
β-Carotene (µg/g)	Fresh	16.20 ^c^	23.74 ^b^	27.18 ^a^	0.223	˂0.001
	Stored	15.55 ^c^	22.95 ^b^	26.21 ^a^	0.108	˂0.001
	SEM ^2^	0.076	0.051	0.083	-	-
	*p*-value ^4^	0.145	0.083	0.066	-	-
Retinol (µg/g)	Fresh	20.43 ^b^	24.65 ^a^	25.13 ^a^	0.124	˂0.001
	Stored	14.75 ^b^	17.21 ^a^	18.54 ^a^	0.096	˂0.001
	SEM ^2^	0.066	0.083	0.054	-	-
	*p*-value ^4^	˂0.001	˂0.001	˂0.001	-	-
α-Tocopherol (µg/g)	Fresh	22.18 ^c^	29.46 ^b^	34.63 ^a^	0.261	˂0.001
	Stored	14.47 ^b^	20.35 ^a^	22.84 ^a^	0.137	˂0.001
	SEM ^2^	0.084	0.107	0.126	-	-
	*p*-value ^4^	˂0.001	˂0.001	˂0.001	-	-
Total phenols (mg GAE/g)	Fresh	1.39 ^b^	2.74 ^a^	2.97 ^a^	0.047	˂0.001
	Stored	1.02 ^b^	1.89 ^a^	1.84 ^a^	0.051	0.008
	SEM ^2^	0.028	0.012	0.009	-	-
	*p*-value ^4^	0.018	0.003	0.007	-	-
Lipid oxidative status of the yolk						
ABTS radical-scavenging activity (µmol TE/g yolk)	Fresh	0.785 ^c^	1.272 ^b^	1.461 ^a^	0.077	˂0.001
	Stored	0.303 ^b^	0.823 ^a^	0.988 ^a^	0.039	˂0.001
	SEM ^2^	0.018	0.022	0.015	-	-
	*p*-values ^4^	˂0.001	˂0.001	˂0.001	-	-
TBARSs (µg MDA/g of yolk)	Fresh	0.812 ^a^	0.578 ^b^	0.511 ^c^	0.045	0.007
	Stored	1.319 ^a^	0.824 ^b^	0.712 ^c^	0.029	˂0.001
	SEM ^2^	0.023	0.015	0.032	-	-
	*p*-value ^4^	˂0.001	0.006	0.004	-	-

^1^ C: standard diet; HT: diet containing 20% HSC and 4% DT (dried whole tomato); HTP: diet containing 20% HSC and 4% DTP (dried tomato pomace). ^2^ SEM: standard error of the mean. ^3^ Effect of diets. ^4^ Effect of storage. TBARS: Thiobarbituric acid reactive substance; MDA: malondialdehyde. ^a–c^ Means within a row with different superscripts are different (*p* < 0.05).

## Data Availability

The data presented in this study are available on request from the author.

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
