# Peer review of "Effect of Dietary Cold-Pressed Hempseed Cake Supplemented with Tomato Waste on Laying Hen Performance and Egg Yolk Lipid Profile and Antioxidant Status Before and After Storage"

_animals, 2024, doi:10.3390/ani14233444_

Round 1
Reviewer 1 Report
Comments and Suggestions for Authors
The manuscript entitled “Effect of Dietary Cold-Pressed Hempseed Cake Supplemented with Tomato Waste on Laying Hens Performances, Lipid Profile and Antioxidant Status of Egg Yolk Before and After Storage” considered the use of tomato by-products/waste in poultry feed, with the possibility of obtaining eggs with higher nutritional quality and health-promoting properties while reducing the environmental footprint. This is a reliably prepared manuscript in which the authors emphasize both the correct and concise presentation of the research topic as the well as a mostly proper attention to the presented methodological details (for example: not only a diet formulation based on tabular data, but also their own analyses), and the subsequent presentation of the results and their detailed discussion.
Introduction:
I do not have any major comments regarding the Introduction section, since it is written typically for the presented research topic and clearly specifies the aim of the research conducted.
L60-63: I suggest replacing the word chickens with laying hens, because chickens are associated with broiler chickens.
Material and methods:
The Materials and methods section is mostly prepared very well. For example, the formulation of diets was made based on the authors’ analyses (L206-210) which is much more precise and accurate than the use of NRC tabular data, etc. Most of the other methods and protocols are also described with adequate detail and references, which makes reading this manuscript pleasant, especially in the case when selected methods, the scope and comprehensiveness of the analyzes taken altogether confirm that the authors have considered the researched concepts in a thoughtful way.
However, I have to point out one major inconsistency in relation to the determination of antioxidant activity using ABTS assay, which must be corrected. These concern the nomenclature and terminology of this method, which is discussed in different ways throughout the manuscript, what is confusing, i.e.:
L131: “and total antioxidant capacity (TEAC: Trolox equivalent antioxidant capacity).”
L258: “2.7.5. Lipid Oxidative Status”
L265: “(ABTS) method”
L311: “TEAC: trolox equivalent antioxidant capacity; TE: trolox equivalent”.
L456: “total antioxidant capacity (TAC)”
L480: “TAC”
L523-524: “and a higher antioxidant capacity (TAC)”
Table 2: “TEAC (μmol TE/100 g)”
Table 9: Total antioxidant capacity (TAC) (μmol TE/g yolk)
For this reason, the authors should specify the method used in the methodology, in the description of the results and discussion, and in the Tables, especially because:
- The ability to reduce free radicals (ABTS) was assessed, which include only the one of the mechanisms of antioxidant action and does not give a complete view about the antioxidant capacity of the tested additives. Therefore, the most correct and reliable name for the assessed antioxidant capacity using ABTS is the term "radical scavenging activity/ability, etc." expressed as TEAC or TE.
- TEAC/TE is the unit of the antioxidant activity and not a fully fixed method protocol. Trolox is commonly used with other assays, like ABTS and FRAP. It is important to clarify this because different methods (FRAP, ABTS, DPPH) may give different sequences of activity when several sources of antioxidants are compared. Therefore, I find it difficult to agree that the DPPH or FRAP test result could also be called or treated as TAC. Even if a very similar T-AOC parameter (performed using dedicated KITs) is often used in the literature, it is based mainly on the ferric reducing ability method.
Also, for the same reason describing the ABTS assay cannot be included in the section “2.7.5. Lipid Oxidative Status” and should be separated. The determined antiradical activity may explain/justify why the MDA levels were lower, but does not provide any information on the degree of lipid oxidation itself.
L220: If the specific AOAC procedure number is given, it is worth completing it also for other analyses given in the section “2.7.1. Proximate Chemical Composition of Feed” (L206-210).
Discussion:
The Discussion includes mostly a proper explanation of the observed differences between treatments, justifying this with comprehensive results of the analyzed traits, and the confrontation with results of other authors. For example: if a feed additive rich in antioxidants was used (tomato wastes), the content of key antioxidant components (lycopene, tocopherols, phenols, etc.) were analyzed and the antioxidant activity/radical scavenging ability (ABTS, TEAC/TE) was evaluated. Going further, by determining the changes in the concentration of antioxidants and fatty acids in the yolk and combining them with the evaluated MDA parameter, it was possible to reveal whether and why the level of changes in yolk lipid peroxidation changes were observed -> all these parts are well connected to each other. I merely suggest that the authors could mention the mechanism of neutralization of free radical as a potential action in limiting peroxidation in egg yolk lipids. This should match the part given in the L739-744.
The description and discussion of the results are very extensive and detailed, and the conclusions, findings, and observations present in the manuscript are supported also by the results of our own research, not just by dry information from the literature. It is worth pointing out that the authors considered also the limitations related to crude fiber in the used additives.
Comments on the Quality of English LanguageL16-20: please rephrase this sentence, split it into 2, etc., because it is very difficult to read and understand the full content.
L323-324: “In addition, DTP had higher concentrations (p<0.05) and total phenols and α-tocopherol but also a higher antioxidant activity” -> I don't fully understand which concentrations are being considered in this part. Please, verify this sentence.
Author Response
|
Thank you very much for taking the time to review this manuscript. The authors accepted all comments. Please find the detailed responses below and the corresponding revisions/corrections highlighted/in track changes in the re-submitted files. Your comments helped us to improve the paper. |
|
Questions for General Evaluation: The manuscript entitled “Effect of Dietary Cold-Pressed Hempseed Cake Supplemented with Tomato Waste on Laying Hens Performances, Lipid Profile and Antioxidant Status of Egg Yolk Before and After Storage” considered the use of tomato by-products/waste in poultry feed, with the possibility of obtaining eggs with higher nutritional quality and health-promoting properties while reducing the environmental footprint. This is a reliably prepared manuscript in which the authors emphasize both the correct and concise presentation of the research topic as the well as a mostly proper attention to the presented methodological details (for example: not only a diet formulation based on tabular data, but also their own analyses), and the subsequent presentation of the results and their detailed discussion.
Comment: The English could be improved to more clearly express the research. Response and Revisions: Agreed. The manuscript was edited in English by MDPI (ID: english-86146). In addition, the manuscript was reviewed by a native English speaker.
Introduction:
I do not have any major comments regarding the Introduction section, since it is written typically for the presented research topic and clearly specifies the aim of the research conducted.
Comment: L60-63: I suggest replacing the word chickens with laying hens, because chickens are associated with broiler chickens. Response and Revisions: Agreed. I, accordingly, modified. I have modified in all manuscript.
Material and methods:
The Materials and methods section is mostly prepared very well. For example, the formulation of diets was made based on the authors’ analyses (L206-210) which is much more precise and accurate than the use of NRC tabular data, etc. Most of the other methods and protocols are also described with adequate detail and references, which makes reading this manuscript pleasant, especially in the case when selected methods, the scope and comprehensiveness of the analyzes taken altogether confirm that the authors have considered the researched concepts in a thoughtful way. However, I have to point out one major inconsistency in relation to the determination of antioxidant activity using ABTS assay, which must be corrected. These concern the nomenclature and terminology of this method, which is discussed in different ways throughout the manuscript, what is confusing, i.e.:
Response and Revisions: We acknowledge your observation and thank you for your support that helped improve our manuscript. We agree that ABTS does not provide a complete picture of the antioxidant capacity of the tested additives, and TEAC/TE is the unit of antioxidant activity. Therefore, we agree with your suggestion and used "ABTS radicals scavenging activity" expressed as TE.
Comment: L131: “and total antioxidant capacity (TEAC: Trolox equivalent antioxidant capacity).” Response and Revisions: Agreed. I, accordingly, modified. …..and ABTS radicals scavenging activity, expressed in TE (Trolox Equivalent).
Comment: L258: “2.7.5. Lipid Oxidative Status” Response and Revisions: Agreed. I, accordingly, modified. 2.7.5. ABTS Radical Scavenging Assay
Comment: L265: “(ABTS) method” Response and Revisions: Agreed. I, accordingly, modified. ……the 2,2-azino-bis(3-etilbenzotiazolin)-6-sulfonic acid (ABTS) assay, ….. The ABTS radical scavenging assay is used to measure the capacity of compounds to by reducing the dark blue ABTS•+ radical cation to colorless ABTS, which can be quantified spectrophotometrically.
Comment: L311: “TEAC: trolox equivalent antioxidant capacity; TE: trolox equivalent”. Response and Revisions: Agreed. I, accordingly, modified. TEAC: trolox equivalent antioxidant capacity; TE: Trolox Equivalent.
Comment: L456: “total antioxidant capacity (TAC)” Response and Revisions: Agreed. I, accordingly, modified. ABTS radicals scavenging activity
Comment: L480: “TAC” Response and Revisions: Agreed. I, accordingly, modified. ABTS radicals scavenging activity
Comment: L523-524: “and a higher antioxidant capacity (TAC)” Response and Revisions: Agreed. I, accordingly, modified. …….and a higher ABTS radicals scavenging activity (p˂0.05)….
Comment: Table 2: “TEAC (μmol TE/100 g)” Response and Revisions: Agreed. I, accordingly, modified. ABTS radicals scavenging activity (µmol TE/100 g)
Comment: Table 9: Total antioxidant capacity (TAC) (μmol TE/g yolk) Response and Revisions: Agreed. I, accordingly, modified. ABTS radicals scavenging activity (µmol TE/100 g)
For this reason, the authors should specify the method used in the methodology, in the description of the results and discussion, and in the Tables, especially because:
- The ability to reduce free radicals (ABTS) was assessed, which include only the one of the mechanisms of antioxidant action and does not give a complete view about the antioxidant capacity of the tested additives. Therefore, the most correct and reliable name for the assessed antioxidant capacity using ABTS is the term "radical scavenging activity/ability, etc." expressed as TEAC or TE.
- TEAC/TE is the unit of the antioxidant activity and not a fully fixed method protocol. Trolox is commonly used with other assays, like ABTS and FRAP. It is important to clarify this because different methods (FRAP, ABTS, DPPH) may give different sequences of activity when several sources of antioxidants are compared. Therefore, I find it difficult to agree that the DPPH or FRAP test result could also be called or treated as TAC. Even if a very similar T-AOC parameter (performed using dedicated KITs) is often used in the literature, it is based mainly on the ferric reducing ability method.
Comment: Also, for the same reason describing the ABTS assay cannot be included in the section “2.7.5. Lipid Oxidative Status” and should be separated. The determined antiradical activity may explain/justify why the MDA levels were lower, but does not provide any information on the degree of lipid oxidation itself. Response and Revisions: Agreed. I, accordingly, modified. 2.7.5. ABTS Radical Scavenging Assay 2.7.6. Oxidation Assessment
Comment: L220: If the specific AOAC procedure number is given, it is worth completing it also for other analyses given in the section “2.7.1. Proximate Chemical Composition of Feed” (L206-210). Response and Revisions: Agreed. I, accordingly, modified. …..methods for DM (gravimetric method; method 934.01), crude protein (CP) (N x 6.25; method 954.01; Kjeltec auto 1030; Tecator Instrumente, Sweden) and crude fat (EE) (petroleum ether extraction; method 920.39; SOXTHERM, C. Gerhardt GmbH, Königswinter, Germany).
Discussion:
The Discussion includes mostly a proper explanation of the observed differences between treatments, justifying this with comprehensive results of the analyzed traits, and the confrontation with results of other authors. For example: if a feed additive rich in antioxidants was used (tomato wastes), the content of key antioxidant components (lycopene, tocopherols, phenols, etc.) were analyzed and the antioxidant activity/radical scavenging ability (ABTS, TEAC/TE) was evaluated. Going further, by determining the changes in the concentration of antioxidants and fatty acids in the yolk and combining them with the evaluated MDA parameter, it was possible to reveal whether and why the level of changes in yolk lipid peroxidation changes were observed -> all these parts are well connected to each other. I merely suggest that the authors could mention the mechanism of neutralization of free radical as a potential action in limiting peroxidation in egg yolk lipids. This should match the part given in the L739-744. Response and Revisions: Agreed. I, accordingly, added. The antioxidants present in tomato waste act as free radical scavengers, limiting peroxidation in egg yolk lipids and the production of MDA. Antioxidants can donate a hydrogen atom, causing the direct quenching of ROS (reactive oxygen species) before they can damage fatty acids, or they can block the action of enzymes (for example, xanthine oxidase and protein kinase C) that directly generate O2 [72 ].
Comment: L16-20: please rephrase this sentence, split it into 2, etc., because it is very difficult to read and understand the full content. Response and Revisions: Agreed. I, accordingly, modified. In the present study, including hempseed cake (HSC) in the diet of laying hens as a source of PUFAs, along with tomato waste (dried whole tomato: DT or dried tomato pomace: DTP) improved the yolk color and the content of antioxidant compounds and omega-3 fatty acids in the egg yolk.
Comment: L323-324: “In addition, DTP had higher concentrations (p<0.05) and total phenols and α-tocopherol but also a higher antioxidant activity” -> I don't fully understand which concentrations are being considered in this part. Please, verify this sentence. Response and Revisions: Agreed. I, accordingly, modified. In addition, DTP had higher concentrations total phenols and α-tocopherol but also a higher ABTS radical scavenging activity compared to DT and HSC (p˂0.001) (Table 2).
|

Reviewer 2 Report
Comments and Suggestions for Authors
Comments and Suggestions for Authors
The current manuscript provides detailed information on the performance of laying hens, egg quality, lipid profile, antioxidant content and oxidative stability of egg yolk before and after storage. The aim of this study, which was to investigate the effects of dietary inclusion of hempseed cake as a source of PUFAs together with a natural source of antioxidants such as dried whole tomato or dried tomato pomace on the performance of laying hens, egg quality, lipid profile, antioxidant content and oxidative stability of egg yolk before and after storage, has been achieved. The authors have done a commendable and substantial job in supporting their claims with appropriate methodology. The manuscript is written with legible illustrations.
However, the manuscript could be considered for publication after addressing the following shortcomings.
1. Line 31: Please put “(40 weeks old)” instead of “(aged between 28 and 37 weeks)”.
2. Line 125: It is important to indicate the drying time. Please indicate the drying time.
3. Line 134: Why did you the 40-weeks-old laying hens in this study? Please explain the reason in the manuscript.
4. Line 143: Please put “diet” instead of “diets”.
5. Line 164: Please put “Diets” instead of “Feed”.
6. Line 185: Please put “albumen” instead of “white”.
7. Line 228: Please put “Diet” instead of “Feed”.
8. Table 4: Please put “Diet” instead of “Feed”.
9. Line 361: Please put “p<0.05” instead of “(P<0.05)”.
10. Line 431: Please put “p<0.05” instead of “(P<0.05)”.
Author Response
|
Thank you very much for taking the time to review this manuscript. The authors accepted all comments. Please find the detailed responses below and the corresponding revisions/corrections highlighted/in track changes in the re-submitted files. Your comments helped us to improve the paper. |
|
Questions for General Evaluation: The current manuscript provides detailed information on the performance of laying hens, egg quality, lipid profile, antioxidant content and oxidative stability of egg yolk before and after storage. The aim of this study, which was to investigate the effects of dietary inclusion of hempseed cake as a source of PUFAs together with a natural source of antioxidants such as dried whole tomato or dried tomato pomace on the performance of laying hens, egg quality, lipid profile, antioxidant content and oxidative stability of egg yolk before and after storage, has been achieved. The authors have done a commendable and substantial job in supporting their claims with appropriate methodology. The manuscript is written with legible illustrations.
However, the manuscript could be considered for publication after addressing the following shortcomings.
Comment 1: Line 31: Please put “(40 weeks old)” instead of “(aged between 28 and 37 weeks)”. Response and Revisions: Thanks for the observation. This error was also reported by the other three reviewers. At line 31 it is correct; the error is from line 134. I have corrected accordingly: A total of 96 TETRA SL laying hens (28 weeks old; 1830±77 g body weight).....
Comment 2: Line 125: It is important to indicate the drying time. Please indicate the drying time. Response and Revisions: Agreed. I, accordingly, added. The tomatoes were cut into small pieces (the spoiled portions were removed), after which they were dried in a static oven at 50 0C together with the tomato pomace for 24 h and then ground using a universal hammer mill with 1 mm mesh.
Comment 3: Line 134: Why did you the 40-weeks-old laying hens in this study? Please explain the reason in the manuscript. Response and Revisions: Agreed. I, accordingly, added. A total of 96 TETRA SL laying hens during the peak laying period (28 to 37 weeks of age; initial body weight: 1830±77 g), when egg production is relatively constant were divided into three groups of 32 birds each (8 replicate cages, 4 hens per cage).
Comment 4: Line 143: Please put “diet” instead of “diets”. Response and Revisions: Agreed. I, accordingly, modified.
Comment 5: Line 164: Please put “Diets” instead of “Feed”. Response and Revisions: Agreed. I, accordingly, modified.
Comment 6: Line 185: Please put “albumen” instead of “white”. Response and Revisions: Agreed. I, accordingly, modified.
Comment 7: Line 228: Please put “Diet” instead of “Feed”. Response and Revisions: Agreed. I, accordingly, added. I added "Diet". I also kept "Feed", because the subtitle refers to the analysis of FA from tested feed (HSC, HT and HTP), diets (C, HT and HTP) and egg yolk.
Comment 8: Table 4: Please put “Diet” instead of “Feed”. Response and Revisions: We believe that this change is not justified. FCR: Feed Conversion Ratio. FCR = Feed intake (kg) / Egg production (kg) (Wikipedia).
Comment 9: Line 361: Please put “p<0.05” instead of “(P<0.05)”. Response and Revisions: Agreed. I, accordingly, modified.
Comment 9: Line 431: Please put “p<0.05” instead of “(P<0.05)”. Response and Revisions: Agreed. I, accordingly, modified.
|

Reviewer 3 Report
Comments and Suggestions for Authors
Mierlita et al. present an interesting study regarding the effects of effects of dietary inclusion of cold-pressed hempseed cake combined with tomato waste on laying hens’ productive performance, and the lipid profile and antioxidant status in hens’ egg yolk before and after storage. This manuscript is fall within the scope and aim of Animals journal. Please find my review comments below.
1. Line 51 (Keywords): Tomato waste is missing in the keywords section. Besides, the health index appears to be an irrelevant keyword.
2. Line 134: The age of the laying hens mentioned here contradicts the age range of the laying hens stated in the abstract (Line 31), which spans from 28 weeks to 37 weeks, while in this case it is 40 weeks.
3. Line 140: “hempseed cake (HSC)”: An abbreviation should appear where its full name first appears. Please double check for similar cases.
4. Line 145: Please provide justifications for the used dosage of DT and DTP (4%) or cite references to validate the appropriateness of these additions.
5. Line 166: Unclear description: The precise content of choline in the choline premix; What is the ronozyme W?
6. Line 307: The gross energy (GE) of the three tested feedstuffs are suggested to be provided in Table 2.
7. Line 307 (Table 2): It does not make sense to compare the chemical composition and antioxidant content in HSC with DT or DTP using statistical analysis.
8. Line 365 (Table 5): In the discussion section, kindly provide a detailed explanation for the noticeable decline in the overall weight of the egg subsequent to a storage period of 30 days, specifically focusing on Lines 568-570.
9. Line 396 (Figure 1): The information depicted in Figure 1B is replicated within Table 3.
10. Line 426 (Table 8): Kindly utilize the identical capitalization style as the initial letter of each word in the following terms: Saturation index, Unsaturation Index, Egg lipid quality index, Elongase index, and Thioesterase index, etc.
11. Line 534: Please modify the phrase “between 72 and 92%” to “between 72% and 92%”. Double check this issue throughout the manuscript.
12. Lines 564-567: Please provide more specific explanation regarding the discrepancy.
13. Lines 752-772: The Conclusion section is excessively lengthy and verbose. Therefore, please re-orgainze this section.
Author Response
|
Thank you very much for taking the time to review this manuscript. The authors accepted all comments. Please find the detailed responses below and the corresponding revisions/corrections highlighted/in track changes in the re-submitted files. Your comments helped us to improve the paper. |
|
Questions for General Evaluation: Mierlita et al. present an interesting study regarding the effects of effects of dietary inclusion of cold-pressed hempseed cake combined with tomato waste on laying hens’ productive performance, and the lipid profile and antioxidant status in hens’ egg yolk before and after storage. This manuscript is fall within the scope and aim of Animals journal. Please find my review comments below.
Comment 1: Line 51 (Keywords): Tomato waste is missing in the keywords section. Besides, the health index appears to be an irrelevant keyword. Response and Revisions: Agreed. I, accordingly, modified. I added: tomato waste. I removed: health index
Comment 2: Line 134: The age of the laying hens mentioned here contradicts the age range of the laying hens stated in the abstract (Line 31), which spans from 28 weeks to 37 weeks, while in this case it is 40 weeks. Response and Revisions: Thanks for the observation. I have corrected accordingly: A total of 96 TETRA SL laying hens (28 weeks old; 1830±77 g body weight).....
Comment 3: Line 140: “hempseed cake (HSC)”: An abbreviation should appear where its full name first appears. Please double check for similar cases. Response and Revisions: Agreed. I, accordingly, modified.
Comment 4: Line 145: Please provide justifications for the used dosage of DT and DTP (4%) or cite references to validate the appropriateness of these additions. Response and Revisions: Agreed. I, accordingly, added. Previous studies [15-17] have shown that supplementing the diet of laying hens or Japanese quail with tomato waste at a level of 4% to 6% improved yolk color and yolk content of antioxidant compounds.
Comment 5: Line 166: Unclear description: The precise content of choline in the choline premix; What is the ronozyme W? Response and Revisions: Agreed. I, accordingly, added. Premix contained 180,000 mg/kg of choline chloride. Ronozyme WX is an additive that contains endo‐1,4‐β‐xylanase (EC 3.2.1.8) and ensure 100 FXU (Fungal Xylanase Units)/kg feed.
Comment 6: Line 307: The gross energy (GE) of the three tested feedstuffs are suggested to be provided in Table 2. Response and Revisions: Agreed. I, accordingly, added. In Table 2 we have added the values for gross energy (GE) (kcal/kg DM). Calculated values acording to NRC [20].
Comment 7: Line 307 (Table 2): It does not make sense to compare the chemical composition and antioxidant content in HSC with DT or DTP using statistical analysis. Response and Revisions: Agreed. I agree with your observation. It is true that it does not make sense to compare the chemical composition and antioxidant content of HSC with DT or DTP using statistical analysis, especially since the differences are quite large and obvious. However, in a previously published article the reviewers requested this statistical comparison. They justified this by the fact that when presenting the results, statements like: greater or less than.......only if we have the value of p (p˂ 0.05) cannot be made. Consequently we decided to keep this comparison through statistical analysis.
Comment 8: Line 365 (Table 5): In the discussion section, kindly provide a detailed explanation for the noticeable decline in the overall weight of the egg subsequent to a storage period of 30 days, specifically focusing on Lines 568-570. Response and Revisions: Agreed. I, accordingly, added. Egg weight loss during storage occurs regardless of the stored conditions [43], which is also confirmed in our research. This was due to the biophysical and chemical changes taking place in the egg content from the moment when the eggs are laid. During storage, solvents (water and other gaseous products) from egg is lost through evaporation, and the rate of evaporation is influenced influenced by storage time, temperature, relative humidity and porosity of the shell. During storage, the pH of the egg albumen increases through the release of carbon dioxide as a result of carbonic acid dissociation, which in turn leads to the weakening of bonds within the ovomuccin-lysosome complex. This promotes egg weight loss [43]. The albumin weight loss during storage were to the water and gases passage from the albumen via the shell pores but aslo diffusion into the yolk [44].
Comment 9: Line 396 (Figure 1): The information depicted in Figure 1B is replicated within Table 3. Response and Revisions: Agreed. The comment is correct, the information depicted in Figure 1B is replicated with in Table 3. We chose to present the most important results of our research in graphic form to make them more accessible to the reader (analyzing data from a table is more difficult). In addition, journals (including Animals) encourage the presentation of important results in graphical form (eg the graphical abstract, which is mandatory in some major journals). Consequently, we decided to keep figure 1b, as well as the data in the table (for statistical analysis).
Comment 10: Line 426 (Table 8): Kindly utilize the identical capitalization style as the initial letter of each word in the following terms: Saturation index, Unsaturation Index, Egg lipid quality index, Elongase index, and Thioesterase index, etc. Response and Revisions: Agreed. I, accordingly, modified. Thanks for the comment.
Comment 11: Line 534: Please modify the phrase “between 72 and 92%” to “between 72% and 92%”. Double check this issue throughout the manuscript. Response and Revisions: Agreed. I, accordingly, modified. Thanks for the comment.
Comment 12: Lines 564-567: Please provide more specific explanation regarding the discrepancy. Response and Revisions: Agreed. I, accordingly, added. Differences between the present study and other experiments regarding the effect of introducing tomato waste into the diet of laying hens on egg performance and quality may be due to several factors, such as (1) tomato variety, soil type, tomato cultivation practices, stage baking and climate; (2) the different methods of processing tomatoes, e.g. that for the production of tomato paste, sauces, ketchup or tomato juice; (3) rate of supplementation of laying hens diet with tomato waste; and (4) other factors (composition of diets, age of hens, genetic factors, duration of experiment).
Comment 13: Conclusion: Needs to be shorter and precise, as well as the ideal recommended level of dietary supplement should be mentioned. Response and Revisions: Agreed. I, accordingly, modified. The obtained results indicate that tomato waste (DT and DTP) contains significant amounts of bioactive phytochemicals (carotenoids, phenols and α-tocopherol), with high antioxidant potential. The incorporation of HSC at a level of 20% in the diet of hens as a source of PUFAs and supplementing the diet with 4% tomato waste caused a significant increase in Roche color score of yolk, in correlation with increased deposition of lycopene, lutein and β - carotene in egg yolks. In addition, supplementing the diet of hens with dried tomato waste improved the antioxidant activity of yolk, which ensures the preservation of the quality of eggs enriched with unsaturated FAs during storage for 30 days at 4 0C. The DTP showed a higher antioxidant content and a higher potential to prevent yolk lipid peroxidation than DT. Further studies are needed to see the impact of tomato waste on digestion processes and the immune system of laying hens but also the stability of bioactive compounds during cooking of eggs. |

Reviewer 4 Report
Comments and Suggestions for Authors
Hempseed cake has been recognized as a valuable source of protein, fiber, and essential fatty acids for poultry nutrition. However, the high cost of conventional feed ingredients has led to research exploring alternative feed sources for poultry. One such alternative is tomato waste, which is a byproduct of the tomato processing industry. Tomato waste is rich in antioxidants, fiber, and vitamins, making it a potentially valuable supplement for poultry feed. By supplementing hempseed cake with tomato waste, poultry producers can provide their laying hens with a nutritionally balanced and cost-effective diet. The current study investigated the effects of the dietary inclusion of hempseed cake as a source of PUFAs together with a natural source of antioxidants such as dried whole tomato or dried tomato pomace on the performance of laying hens, egg quality, lipid profile, antioxidant content and oxidative stability of egg yolk before and after storage. It is an interesting article that can add to the poultry nutrition field, despite that the main concept was used in your previous publication “Mierlita et al. 2024” in Animals. However, the following suggestions and comments need to be addressed to improve this manuscript.
L.13-16: Also, give a brief on “Hempseed Cake”.
L.32-33: You mentioned before that you divided them into three groups! Avoid repetition.
L.35: “introduction”: Did you mean “incorporation”?
L.53: “Introduction”: Comprehensive and well written.
L.60: Replace “and” with “with”.
L.65: Remove “but”.
L.65-68: Where is the reference?
L.68: Change “found” to “indicated”.
L.72-73: Rewrite it clearly.
L.95-98: What is your explanation for these confusing findings?
L.134: The hens’ age wasn’t like that mentioned in the abstract!!
L.140-148: This paragraph in its form is not suitable to be included in the materials & methods! Rewrite it and move (or delete) the unneeded statements.
L.161-162: Not clear!
L.176: Provide the number of samples.
L.178: Is it for calculating the “feed consumption” or the “feed intake”?
L.180-181: Cannot well understand it!
L.202-203: Expand how?
L.236: “Schlatter”: The spelling wasn’t compatible with that mentioned in the references’ list! Revise it.
Discussion: What is your scientific explanation of increasing FI as a result of this dietary addition?
And do you think it is a desirable result?
Table 2: How did you deal with the high level of fiber, especially for DTP?
Table 4: Correct to “Average egg weight (g)”. And compare this row’s values with the “egg weight (g)” values of table 5.
L.721: “the studies” or “the study”?
Table 6: Carefully check the “C16:1” values’ significance and “C18:3 n-3” values!
Table 7: “Total MUFA” and “Total PUFA” rows: Show the significance for the fresh eggs because the results are significant.
Same previous comment for the other next items in the table.
Table 9: Please revise the “Lutein (μg/g)” values’ significance letters!
L.445: “food byproducts”! Did you mean the “dried whole tomato and dried tomato pomace”? Please be accurate.
L.493-494: Clarify.
Conclusion: Needs to be shorter and precise, as well as the ideal recommended level of dietary supplement should be mentioned.
Author Response
|
Thank you very much for taking the time to review this manuscript. The authors accepted all comments. Please find the detailed responses below and the corresponding revisions/corrections highlighted/in track changes in the re-submitted files. Your comments helped us to improve the paper. |
|
Questions for General Evaluation: Hempseed cake has been recognized as a valuable source of protein, fiber, and essential fatty acids for poultry nutrition. However, the high cost of conventional feed ingredients has led to research exploring alternative feed sources for poultry. One such alternative is tomato waste, which is a byproduct of the tomato processing industry. Tomato waste is rich in antioxidants, fiber, and vitamins, making it a potentially valuable supplement for poultry feed. By supplementing hempseed cake with tomato waste, poultry producers can provide their laying hens with a nutritionally balanced and cost-effective diet. The current study investigated the effects of the dietary inclusion of hempseed cake as a source of PUFAs together with a natural source of antioxidants such as dried whole tomato or dried tomato pomace on the performance of laying hens, egg quality, lipid profile, antioxidant content and oxidative stability of egg yolk before and after storage. It is an interesting article that can add to the poultry nutrition field, despite that the main concept was used in your previous publication “Mierlita et al. 2024” in Animals. However, the following suggestions and comments need to be addressed to improve this manuscript.
Comment: L.13-16: Also, give a brief on “Hempseed Cake”. Response and Revisions: Agreed. I, accordingly, added. Also, hempseed cake (HSC) has been recognized as a valuable source of protein and essential fatty acids for poultry nutrition.
Comment: L.32-33: You mentioned before that you divided them into three groups! Avoid repetition. Response and Revisions: Agreed. I, accordingly, modified. A total of 96 laying hens (aged between 28 and 37 weeks) were divided into three groups of 32 birds each (eight replicate cages, four birds per cage) and were assigned randomly the following dietary treatments: a standard corn–soybean meal diet (C), a diet containing 20% hempseed cake and 4% dried whole tomato (HT) and a diet containing 20% hempseed cake and 4 % dried tomato pomace (HTP).
Comment: L.35: “introduction”: Did you mean “incorporation”? Response and Revisions: Agreed. I, accordingly, modified. I have replaced "introduction" with "incorporation" throughout the manuscript.
Comment: L.53: “Introduction”: Comprehensive and well written. Response and Revisions: Thanks for the appreciation.
Comment: L.60: Replace “and” with “with”. Response and Revisions: Agreed. I, accordingly, modified.
Comment: L.65: Remove “but”. Response and Revisions: Agreed. I, accordingly, modified.
Comment: L.65-68: Where is the reference? Response and Revisions: Agreed. I have added the much needed references.
Comment: L.68: Change “found” to “indicated”. Response and Revisions: Agreed. I, accordingly, modified.
Comment: L.72-73: Rewrite it clearly. Response and Revisions: Agreed. I, accordingly, added. An alternative to synthetic additives with an antioxidant role used in poultry feed could be plants and food waste rich in antioxidant compounds.
Comment: L.95-98: What is your explanation for these confusing findings? Response and Revisions: Agreed. Differences between the present study and other experiments regarding the effect of introducing tomato waste into the diet of laying hens on egg performance and quality may be due to several factors, such as (1) tomato variety, soil type, tomato cultivation practices, stage baking and climate; (2) the different methods of processing tomatoes, e.g. that for the production of tomato paste, sauces, ketchup or tomato juice; (3) rate of supplementation of laying hens diet with tomato waste; and (4) other factors (composition of diets, age of hens, genetic factors, duration of experiment). This explanation was introduced in the Discussions chapter (4.2. Performance of Laying Hens and Egg Quality).
Comment: L.134: The hens’ age wasn’t like that mentioned in the abstract!! Response and Revisions: Agreed. Thanks for the observation. I have corrected the error.
Comment: L.140-148: This paragraph in its form is not suitable to be included in the materials & methods! Rewrite it and move (or delete) the unneeded statements. Response and Revisions: Agreed. I, accordingly, modified. In the present study, HSC was used to enrich the egg yolk with n-3 FA at the inclusion level of 20% in the hens' diet in agreement with a previous study [6]. The DT and DTP were included in the diet of laying hens at a level of 4%, along with HSC, to serve as complementary sources of natural antioxidants.
Comment: L.161-162: Not clear! Response and Revisions: Agreed. I, accordingly, added. To ensure the same content of CP (crude protein) and ME (metabolizable energy) in all diets, in the standard diet (C) we incorporated sunflower meal and sunflower oil, respectively (Table 1).
Comment: L.176: Provide the number of samples. Response and Revisions: Agreed. I, accordingly, added. Feed intake, the number of eggs and their weight were recorded separately for each replicate (n=8 replicates/treatment).
Comment: L.178: Is it for calculating the “feed consumption” or the “feed intake”? Response and Revisions: Agreed. I, accordingly, modified. I replaced "feed consumption" with "feed intake".
Comment: L.180-181: Cannot well understand it! Response and Revisions: Agreed. I, accordingly, added. Feed intake, the number of eggs and their weight were recorded separately for each replicate (n=8 replicates/treatment). Each replicate was housed in a cage (4 laying hens/cage). For example, eggs were harvested daily from each cage in a treatment: 8 cages x number of eggs/cage = number of eggs/treatment.
Comment: L.202-203: Expand how? Response and Revisions: Agreed. I, accordingly, added. For yolk cholesterol analysis, 24 eggs/treatment (3 eggs/replicate) were sampled during the last week of the experiment.
Comment: L.236: “Schlatter”: The spelling wasn’t compatible with that mentioned in the references’ list! Revise it. Response and Revisions: Agreed. I, accordingly, modified. “Schlatterer”
Comment: Discussion: What is your scientific explanation of increasing FI as a result of this dietary addition? Response: The increase in feed intake in hens of the HT and HTP groups could be related to the presence of ascorbic acid in dried tomato waste, which increases appetite and therefore feed intake [Khan et al., 2023]. According to Colombino et al. [2020], the increase in feed intake could be related to the high fiber content of dried tomato pomace and hempseed cake, which accelerated the speed of feed transit in the intestine, reducing digestion and absorption of nutrients. This could explain the lack of significant differences compared to the control group regarding egg production, egg weight and FCR.
Comment: And do you think it is a desirable result? Response: The increasing FI as a result of this dietary addition it is not desirable, because egg production has not increased (could lead to higher expenses).
Comment: Table 2: How did you deal with the high level of fiber, especially for DTP? Response: The high fiber content of these dietary addition, especially for DTP, probably reduced the digestion and absorption of nutrients from the feed, and the birds increased their FI to compensate for the lack of nutrients. In addition, even if the FI increased, the decrease in digestion and absorption of nutrients did not allow the improvement of productive performances.
Comment: Table 4: Correct to “Average egg weight (g)”. And compare this row’s values with the “egg weight (g)” values of table 5. Response and Revisions: Agreed. I, accordingly, modified in table 4 and table 5.
Comment: L.721: “the studies” or “the study”? Response and Revisions: Agreed. I, accordingly, modified: “the study”.
Comment: Table 6: Carefully check the “C16:1” values’ significance and “C18:3 n-3” values! Response and Revisions: Agreed. We have carefully checked the meaning of the "C16:1" values and the "C18:3 n-3" values in Table 6 and do not require changes.
Comment: Table 7: “Total MUFA” and “Total PUFA” rows: Show the significance for the fresh eggs because the results are significant. Same previous comment for the other next items in the table. Response and Revisions: Agreed. I, accordingly, added. I have added the omitted meanings.
Comment: Table 9: Please revise the “Lutein (μg/g)” values’ significance letters! Response and Revisions: Agreed. I, accordingly, modified. I revised the meaning letters and changed to the correct form.
Comment: L.445: “food byproducts”! Did you mean the “dried whole tomato and dried tomato pomace”? Please be accurate. Response and Revisions: Agreed. I, accordingly, added. Table 9 shows the effects of food byproducts (HSC, DT and DTP) added to the diet of laying hens………..
Comment: L.493-494: Clarify. Response and Revisions: Agreed. I, accordingly, added. PC2 represented 5.70% of the variation and was positively correlated with cholesterol and negatively with linoleic acid.
Comment: Conclusion: Needs to be shorter and precise, as well as the ideal recommended level of dietary supplement should be mentioned. Response and Revisions: Agreed. I, accordingly, modified. The obtained results indicate that tomato waste (DT and DTP) contains significant amounts of bioactive phytochemicals (carotenoids, phenols and α-tocopherol), with high antioxidant potential. The introduction of HSC at a level of 20% in the diet of hens as a source of PUFAs and supplementing the diet with 4% tomato waste caused a significant increase in Roche color score of yolk, in correlation with increased deposition of lycopene, lutein and β - carotene in egg yolks. In addition, supplementing the diet of hens with dried tomato waste improved the antioxidant activity of yolk, which ensures the preservation of the quality of eggs enriched with unsaturated FAs during storage for 30 days at 4 0C. The DTP showed a higher antioxidant content and a higher potential to prevent yolk lipid peroxidation than DT. Further studies are needed to see the impact of tomato waste on digestion processes and the immune system of laying hens but also the stability of bioactive compounds during cooking of eggs. |
